# Evaluation of Comfort Models Considering the Peculiarities of Hospitalization: Bedding, Clothing and Reduced Activity of Patients

**Silvia Ruggiero** [1] , **Francesco Tariello** [2,*] **and Giuseppe Peter Vanoli** [3]

1 DING–Department of Engineering, University of Sannio, 82100 Benevento, Italy; sruggiero@unisannio.it
2 Department of Agricultural, Environment and Food Sciences, University of Molise, 86100 Campobasso, Italy
3 Department of Medicine and Health Sciences–Vincenzo Tiberio, University of Molise, 86100 Campobasso, Italy; giuseppe.vanoli@unimol.it
* Correspondence: francesco.tariello@unimol.it

**Abstract:** The study of thermo-hygrometric comfort in hospitals involves several factors: the presence of different subjects: patients, operators, visitors; different conditions of hospitalization: patients bedridden or out of bed; psychological aspects and therapeutic treatments. In this paper, the analysis focuses on patients in ordinary hospitalization rooms of a hospital located in southern Italy. Different room orientations, several characteristics, and specific factors concerning hospitalized patients' conditions that significantly influence the comfort indices have been considered. In total, 41 scenarios have been defined and analyzed by means of two comfort models: static and adaptive. The study aims to investigate the application of these models to the complex environment of hospitals, finding strengths and weaknesses, which also results in a re-definition of the HVAC system operation. Results show that patient position (in bed or out), clothing type, and level of coverage in the bed can make the same microclimatic condition more suitable for one scenario over another. Furthermore, room exposure has an effect on the comfort of the indoor temperature. The seasonal analyses highlight that during summer, for all scenarios considering bedridden patients, more than 50% of the PMV calculated values are out of the comfort zone. In winter, the indoor conditions are good for bedridden patients with a cover level of 67% during the nighttime (almost 100% PMV values in comfort zone), while during the daytime, they are more suitable for a 48% coverage level if the patient is in bed or if they are walking (lower than 10% dissatisfied).

**Keywords:** patients' thermal comfort; healthcare facilities; PMV index; adaptive approach; nosocomial environmental conditions; clothing thermal resistance

## 1. Introduction

In the last period, in which the World Health Organization (WHO) declared the COVID-19 pandemic, the essential role of hospitals has been accentuated. Comfort within healthcare facilities is essential for successful care both from the point of view of the patients—who receive treatments and benefit from optimal surrounding conditions (microclimate conditions) to increase their effectiveness—and from the point of view of the medical staff that administer the treatments. The PMV (predicted mean vote), after more than 50 years (it was introduced by Fanger in 1970 [1]), is still used in numerous studies concerning comfort in healthcare facilities.

The PMV index consists of a six-parameter model (four objective parameters: air temperature, relative humidity, air speed, and average radiant temperature; and two subjective ones: clothing and degree of activity), which expresses the comfort or discomfort condition on a scale of seven values, from −3 very cold to +3 very hot. Standards commonly adopted in Europe and the United States to assess comfort in buildings, including UNI EN ISO 7730 [2] and ASHRAE 55 [3], are based on PMV.

A recent literature review paper in 2020 [4] identified 62 works consistent with the four research questions concerning thermal comfort in the health sector. Twenty-four of these works are based on the evaluation of the PMV index; even if it was initially introduced for a moderate indoor environment and for people with normal no-compromised health conditions, it was widely applied to the nosocomial theaters. Historically, this index is the most used in studies relating to comfort in environments with different intended uses: schools, offices, industries, etc. Moreover, in the health sector, there are studies concerning different structures, mainly hospitals, but also elderly centers [5,6].

Healthcare facilities are air-conditioned buildings (in some cases only heated) with exclusive requirements connected to medical treatments, indoor air quality, and disease diffusion control, which is aimed to avoid pathogen transport [7]. Thermal comfort in a context with a variety of occupants, staff, visitors, and patients, who have specific needs and sensations on the basis of their problems, is intrinsically difficult to make for everyone simultaneously [8]. Furthermore, in hospitals, attention must be paid to hygiene, bacteria, and virus proliferation [9]. The combination of temperature and humidity affects the diffusion and survival of bacteria and viruses [10]. Their resistance generally depends on micro-climate environmental conditions differently from species to species.

An overview of the most recent papers treating thermo-hygrometric comfort in hospitals is reported here below. Hospitals are among the most complex types of buildings, gathering different functions, operating 24 h a day, 365 days a year, and hosting different wards and room types, e.g., emergency rooms, operating rooms, long- and short-stay rooms, laboratories, clean rooms, isolation rooms. Each of these rooms requires different climatic conditions because the comfort needs of the patients who occupy them are different. The study of comfort sometimes follows a broader approach by connecting, for example, to energy retrofittings [11,12]—hospitals are energy-intensive structures—or in some cases, comfort is linked to Indoor Air Quality or, more generally, to Indoor Environmental Quality [13–15]. In a paper by Wu et al. [16], thermo-hygrometric comfort was also correlated with acoustic comfort. The study analyzed data obtained from 18 Chinese hospitals in the Northern cold region. It was observed that the temperatures recorded in the rooms were on average higher than those established in the project, while the relative humidity was lower. Of the 220 patients (110 male and 110 female) interviewed, 87% said that the environment was comfortable or very comfortable. Between the three analyzed factors, temperature appeared to be the factor with the most influence on comfort; meanwhile, the acoustic aspects were less relevant, and the variation in relative humidity was insignificant. Patients demonstrated comfort sensations for temperatures between 26 °C and 28 °C and a relative humidity of 40%. With higher temperature levels, patients accepted noisier environments but were more susceptible to sounds.

In the work of Aloitabi and Lo [17], 522 patients agreed to participate in the research carried out in two hospitals of Saudi Arabia: a state-of-the-art private hospital and a public, specialist one. In the first structure, patients with specific pathologies of the cardiology and oncology wards were considered; in the second, the patients had different pathologies and were hospitalized in the surgery and medical wards. The thermal sensation vote (TSV) expressed by patients was between −1 and +1 in 72% and 67% of the answers, respectively, in the two hospitals indicated above; about 70% of patients said they were in a comfortable condition. The analysis of relative humidity showed that in almost all cases, it was less than 60% and, on average, less than 40% in all 18 rooms considered; it also did not seem to be related to temperature. The analysis of the data collected leads to the conclusion that patients perceive warmer conditions during their hospitalization period and that this perception is not attributable to the change in temperature at a certain time of day, nor to the change in relative humidity.

The air-conditioning of hospitals is complicated due to conflicting needs. From the data collected in situ and reported in [17], it can be observed that: in the surgery ward, the trend of the temperatures chosen in the different rooms were approximately the same, and the most recurrent values were between 22.2 °C and 23.9 °C; in the medical ward,

the temperature trend showed two more frequent values; in the oncology rooms, the desired temperatures were, on average, higher than in the other wards; for cardiology, the most recurrent values were less than 22 °C. From this situation, the Authors of the research derived that it is not possible to uniquely define a comfort zone for temperatures that vary too much, and it is necessary to modify the conventional model of the PMV. However, the same Authors say that the PMV index has, in some cases, proved to be adequate for a nosocomial setting [18], while in some others, it has not, as it overestimates or underestimates thermal sensations [19–22].

Alotaibi et al. [19] considered opinions about the thermo-hygrometric conditions of 120 patients of the surgical and medical wards of the International Medical Centre in Jeddah, Saudi Arabia. Data collected during the summer were compared with the PMV of field measurements (carried out from 11:00 to 16:00) when no doctors or visitors stayed with patients. The mean value for the PMV was about −0.5, whereas the mean TSV was 0.32, highlighting a significant discrepancy between the two distributions. The Fanger's model underestimates the real patients' sensations due to the light clothes used and the reduced metabolic rate. The temperatures for which the PMV and TSV are equal to zero can be extrapolated from the analyses and are equal to about 25.5 °C and 22.5 °C, respectively. Furthermore, Azizpour et al. [20] confirmed that patients in large-scale hospitals in hot-humid areas would prefer cooler environmental temperatures with respect to the neutral temperature.

In [21], the applicability of the PMV model in the hospital setting and the suitability of the standard conditions for the tropical conditions of Bangkok were verified. The analyses were carried out by collecting experimental data in two hospitals involving a total of 928 occupants (451 patients, 331 visitors, and 146 staff members). The medical staff expressed a sensation of cold while, on the basis of the PMV (which is equal to 1), the indoor environment was slightly warm. On the other hand, the most common vote among visitors and patients was zero. The temperature ranges that determine less than 20% dissatisfied are: 21.8 °C–27.9 °C for patients, 22 °C–27.1 °C for visitors, and 24.1 °C–25.6 °C for medical staff; however, according to the Thai standard, the comfort range should be 20 °C–25 °C. By considering the obtained results, it emerges that the PMV model is not suitable for hospitals, especially far-from-neutrality temperatures, or for the staff. Patients accept a wider range of conditions due to their diseases and low expectations.

From the data collected in eight hospital wards located in a central region of Italy, del Ferraro et al. [22] compared the PMV and the judgments of 30 patients and 19 medical staff operators. Furthermore, the Authors studied the effect of age and gender on thermal sensation and verified that the PMV index is able to take these effects into account through parametric and non-parametric tests. The TSV distribution of patients was concentrated around zero, whereas that of staff was shifted towards warm. The distribution of patients' answers was tight, even if age differences were considered. The PMV index was not a good predictor of patients' sensation regarding the indoor climate; it is better for medical staff and more adequately represents male responses than females'. Furthermore, the best correlation appeared with the data of staff under 65 years.

Angelova and Velichkova [23] highlighted, throughout their assessment of PMV-PPD (predicted percentage of dissatisfied) indices, the different comfort sensations of doctors and patients in an operating room. They compared alternative clothes and climatic conditions and observed that a condition does not exist that can simultaneously satisfy all of the main occupants of the operating theatre. The lowest percentage dissatisfied (48% of patients and 44% of doctors) took place at 28 °C and a relative humidity of 30%, but this condition causes thermal stress for surgeons.

To overcome the lack of a regulatory framework relating to comfort and air quality in building environments on Madagascar Island, an experimental and subjective study was presented in [24]. Five large hospitals, naturally ventilated, were analyzed. A total of 400 people were involved in the surveys and interviews during both the rainy and dry seasons. The approach followed in this study was based on both PMV and adaptive

comfort, taking into account the influence of gender, clothing, activities, mind state, and control strategies of the occupants. By comparing the responses of males and females regarding thermal satisfaction, thermal preference, and thermal comfort, no obvious results were observed in both types of buildings considered. From the statistical analysis, it could be seen that at 26.41 °C, an equal number of people would like a warmer and colder environment; for this reason, it was assumed to be the preferred temperature. According to the adaptive approach, when the outdoor temperature varies between 20.3 °C and 29.5 °C, the corresponding indoor comfort temperature is in the range of 23.4 °C–26.8 °C, and a good linear interpolation of the data appears. The neutral operating temperature varies between males and females, with the former being 24.2 °C, and the latter, 23.65 °C. In hospitals, there is a percentage of dissatisfaction lower than 10% in the temperature range 22.4 °C–25.3 °C, but 99% of hospitalized patients find the indoor microclimate comfortable.

The work of Derks et al. [25] complained of a limited number of studies concerning nurses' comfort and was developed to reduce this gap. The data were collected in two wards of a hospital in The Netherlands during the summer and autumn period of 2016 to verify the perception of the microclimate during the working hours and the impact it has on the performance of the caretakers. The thermo-hygrometric conditions currently maintained in the wards are slightly unacceptable by the nurses and, in part, affect their performance; therefore, it is assumed that there is a need to maintain different climatic conditions in the different areas of the hospitals. The results show that the correlation between temperature and the TSV or thermal acceptance vote is linear even if the link with TSV is less marked; this means that as the temperature increases, the nurses tend to express that the sensation is unacceptably warm. With respect to work performance, most of the answers revealed obstructive conditions to adequately carrying out the required tasks.

Khalid et al. [26] evaluated the existing comfort conditions and preferences of patients and visitors in Malaysian hospitals. In particular, the responses of 305 patients and 84 visitors were collected in 3 hospitals in Kuala Lumpur (wards of medicine, surgery, maternity, and pediatrics). The data collected showed that 65% of patients and 55% of visitors fell within the range (−1, +1). Over half of the visitors and patients voted in the negative part of the scale, and their mean votes were −1.1 and −0.9, respectively. The judgment relative to humidity was very often (in more than 60% of cases) in the neutrality interval and was symmetrically distributed around zero. Air movement was acceptable for 86% of the occupants if the operating temperature remained between 22 °C and 28 °C. From the analysis of the data, it was further observed that the parameter that most affects thermal preference, overall comfort, and judgment of air quality of the patients is their state of health. State of health was more influential than age, gender, or hospitalization days.

The thermal responses of the body (temperature regulation, vasoconstriction, and vasodilatation), metabolic rate, and clothing can be significantly altered in hospitalized people with respect to healthy ones due to: the pharmacological treatments they are subject to, the diseases they have, the expectation about the surgical operation they are going to receive, and the place where they spend most of the day (patients lay in bed). In [27], the PMV model has been adapted and improved to take into account peculiarities of the inpatients' stay in the healthcare facilities. Modified equations for the metabolism and the clothing thermal resistance have been introduced in the standard model. Experimental data validation confirmed an excellent correspondence between numerical results and in-bed patient responses during the winter period (5% difference). By solving the modified model, typical comfort charts (operative–wet bulb temperature) have also been elaborated on for general hospitals wards.

A dilemma emerging from thermal comfort studies in health facilities is that the use of common thermal comfort indices, conventionally applied in other buildings, is not always suitable to represent and understand patients' behaviors and needs that are conditioned by the disease level, the state of health, medical aspects, therapeutic treatments, and further potential factors [17].

However, PMV is the index most often used for the assessment of comfort in the hospital setting, and it can provide an appropriate representation of patients' comfort status if suitably modified. In most papers, only temperature and humidity are measured (PMV depends on four objective variables), and often, PMV is applied over its validation range. Therefore, the aim of this paper is to evaluate whether thermo-hygrometric conditions conventionally maintained in patients' rooms of a hospital in southern Italy can be considered adequate by applying the standard comfort model of PMV, taking into account the modified thermal insulation of clothes, bed staying, and a reduced metabolic rate. A multitude of combinations is evaluated, considering different patient positions (in bed and out of bed), different types of clothing, and alternative solutions for bedding. The thermal resistance considered is calculated as the sum of clothing and bed resistance. For both terms, different configurations (different percentages of cover, alternative types of beddings, and clothes) are considered. For the metabolic rate, three possible levels are established (sleeping patients, laying/sitting patients, standing/walking patients).

These evaluations are carried out over four standard days, one per season, and simulated data is also statistically treated for the four entire seasons, considering patients without acute medical conditions. Furthermore, the adaptive model of ASHRAE Standard 55 is also assessed.

## 2. Case Study

The building chosen for this study was the "Ferdinando Veneziale" Hospital. It is located in the city of Isernia, an Italian backcountry, characterized by 1866 Heating Degrees-Day (baseline 20 °C). According to the Köppen climate classification, Isernia has a hot-summer Mediterranean climate (Csa).

The hospital was built between 1976 and 1986, but over the years, it has undergone several renovations, expansions, and maintenance works. The building, with a gross volume of about 88,000 m$^3$, has an articulated geometry, arranged on 2 underground and 6 aboveground floors. The main facade is along the east-west direction (Figure 1).

An in-depth energy audit of the building in question was presented in a previous study [28]. In situ surveys, interviews, and measurements were used for characterizing the envelope-HVAC system.

The opaque building envelope is composed of two main types of walls: one made by masonry and one by prefabricated concrete panels. The thickness of both walls is 0.50 m overall, with an average thermal transmittance of 1.58 W/m$^2$K. The horizontal elements of the basement floor are made of concrete beams, joists, and hollow bricks without insulating layers; the roof has the same structure, but with the presence of a light insulating layer. The U-value is $\approx$ 1.80 W/m$^2$K and $\approx$ 0.70 W/m$^2$K for the basement floor and roof, respectively. The transparent envelope has an incidence on the wall of about 16%. It is made of a single clear glass and metallic frame with an overall window thermal transmittance ($U_W$) of 4.15 W/m$^2$K.

The composition of the HVAC system includes heating systems with radiators in hospital wards, all-air systems for all surgery blocks and rooms, and a mixed air/water HVAC system, given by the combination of radiators and air handling units in corridors, ambulatories, and offices. The heating system is composed of two centralized boilers, also used for providing DHW, with a total thermal capacity of 7000 kW and nominal efficiency of 96%. Moreover, 11 air handling units are installed. For the cooling service, there are autonomous split systems installed in a few rooms and two water-cooled chillers (cooling capacity of 1860 kW each). The heating period goes from 1 November to 15 April with a continuous operating schedule (24 h) at the set-point temperature of 20 °C. The cooling period was assumed to be between 1 May to 30 September (10 h every day) at the set-point temperature of 26 °C. This is the real heating/cooling schedule used in the building, obtained by consulting the technical/maintenance office. Finally, as a lighting system, fluorescent lamps were used.

Further information regarding other dimensions and geometrical peculiarities, as well as the HVAC system and its real operation, are shown in Table 1.

**Table 1.** Main characteristics of the whole building.

| Main Building's Dimensions and Geometry | | | | |
|---|---|---|---|---|
| Total building area: 27,342.80 m$^2$ | | | Gross roof area: 8200.97 m$^2$ | |
| Maximum height: 18.87 m | | | Total volume: 88,100.38 m$^3$ | |
| Net conditioned building area: 24,474.15 m$^2$ | | | Conditioned volume: 78,821.25 m$^3$ | |
| Building envelope | | | | |
| External wall U = 1.58 W/m$^2$K | Basement floor U = 1.80 W/m$^2$K | | Window U = 4.15 W/m$^2$K | |
| Internal wall U = 1.81 W/m$^2$K | Roof U = 0.70 W/m$^2$K | | Solar factor: 0.86 | |
| Heat transfer area of external walls, roof, and fenestration of the examined building | | | | |
| | Total | North | East | South | West |
| Gross wall area [m$^2$] | 10,673.43 | 3282.67 | 1971.24 | 3460.47 | 1959.05 |
| Window opening area [m$^2$] | 1529.61 | 486.56 | 305.77 | 459.50 | 277.78 |
| Above ground window-wall ratio [%] | 16.02 | 17.80 | 16.94 | 13.93 | 16.22 |
| HVAC system and operation | | | | |
| Heating service | 2 boilers | Total thermal capacity: 7000 kW | | Nominal efficiency of 96% |
| DHW service | 4 thermal storages | | Total capacity: 5000 L | |
| Cooling service | 2 water-cooled chillers | | Total cooling capacity: 3720 kW. | |
| Ventilation service | 11 air handling units (not equipped with heat recovery systems) with liquid water humidifiers | | | |
| Operation | Heating period | 1 November–15 April | 24 h | Set-point temperature: 20 °C |
| | Cooling period | 1 May–30 September | 10 h (9:00–19:00) | Set-point temperature: 26 °C |

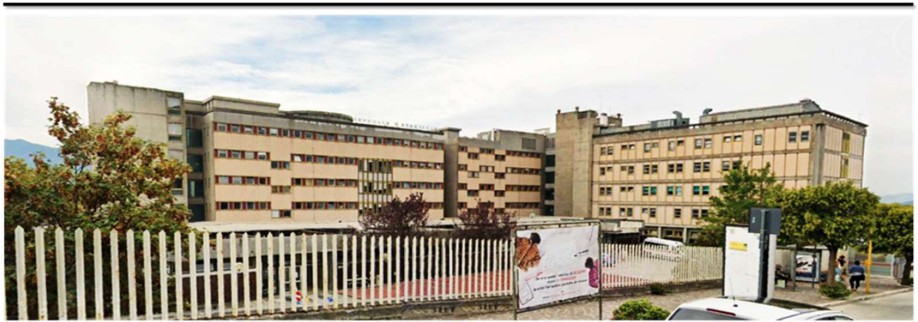

**Figure 1.** "Ferdinando Veneziale" Hospital main entrance.

These data were the starting point in the creation of the numerical model of the hospital in the EnergyPlus engine (developed by U.S. Department of Energy's (DOE) Building Technologies Office (BTO), Washington, DC, USA) [29] by means of its graphical interface DesignBuilder (developed by DesignBuilder Software Ltd., Gloucester, UK) [30]. In this paper, the transient heat transfer through the building was solved by using Conduction Transfer Function algorithms (CTF), while the time steps for the running of the simulations "Number of time-steps per hour" was set equal to 6.

The developed numerical model (Figure 2a) was calibrated by comparing the simulation outputs to the measured energy data (referring to the nine available years). Figure 2b shows the monthly comparison between the real gas consumption and the results of the dynamic energy simulation, as well as the calculated errors (EER$_{month}$). Moreover, the values of other calibration indexes are also shown, such as the error in the annual energy consumption (EER$_{average,year}$), the mean bias error (MBE), and the coefficient of variation of the root mean squared error CV(RMSE$_{month}$). It can be seen that all data were within the threshold range defined by the most accredited and tightened guideline: the Measurement

and Verification [31]. Electricity was done with the same evaluation, with an $ERR_{average,year}$ equal to 7.0%. Therefore, the numerical model developed could be considered as calibrated. It should be stressed that a calibrated model that faithfully describes the energy requests of the real building also simulates its thermal behavior and, therefore, the real indoor microclimate; as was shown by Bellia et al. [32] in a case study located in the same climate zone, and using the same software, as the present one. Calibrating the numerical model according to natural gas consumption and electricity demand, the hourly air temperature profile is comparable with the measured one, with the hourly error always <30% (limit value for hourly data according to [31]). On the basis of this observation, it can be concluded that the model is suitable for patient comfort investigation.

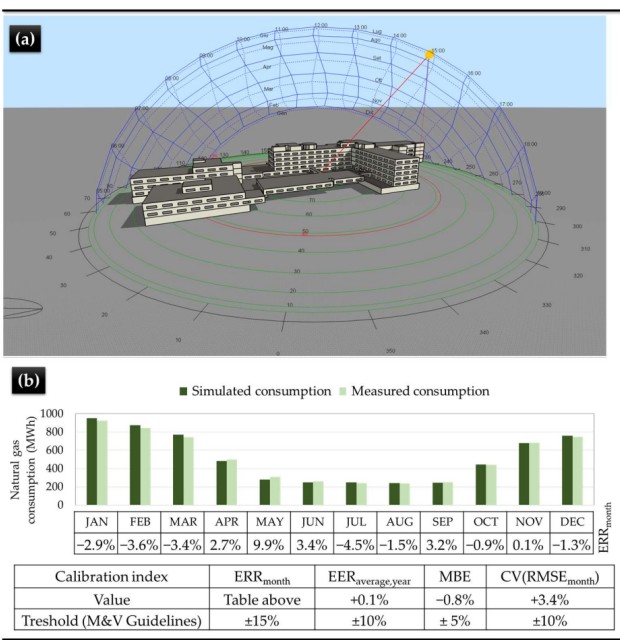

**Figure 2.** Numerical model developed (**a**); comparison between monthly real consumptions and energy simulation results (**b**).

## 3. Methodological Approach

### 3.1. Analyzed Thermal Zones

The hospital ward under investigation was the surgery, located on the second floor of the building. The ward offers services both in ordinary hospitalization and in day-surgery and outpatient. For ordinary hospitalization, there are a total of 11 rooms. The analyzed zone has a very regular geometry oriented along the east-west direction, as shown in Figure 3a. The two chosen rooms, called "Room A" and "Room B", are on opposite exposures and are separated by an internal corridor (Figure 3b). In detail, the two rooms have the same intended use (ordinary hospitalization), similar geometric characteristics, and approximately 33 $m^2$ of floor area (including bathrooms). Similar radiator positions and ventilation duct distributions are also present, as depicted in Figure 3c. The rooms each have a single external vertical wall, with the same percentage of window area. The thermo-physical characteristics of the opaque building envelope are detailed in Table 2, where $t$ is the thickness, $\lambda$ is the thermal conductivity, $\rho$ is the density, $c_p$ is the specific heat, $R$ is the thermal resistance, and $U$ is the thermal conductance. The only difference between the two rooms is the orientation of the external walls: Room A has the facade facing south, while Room B is facing north.

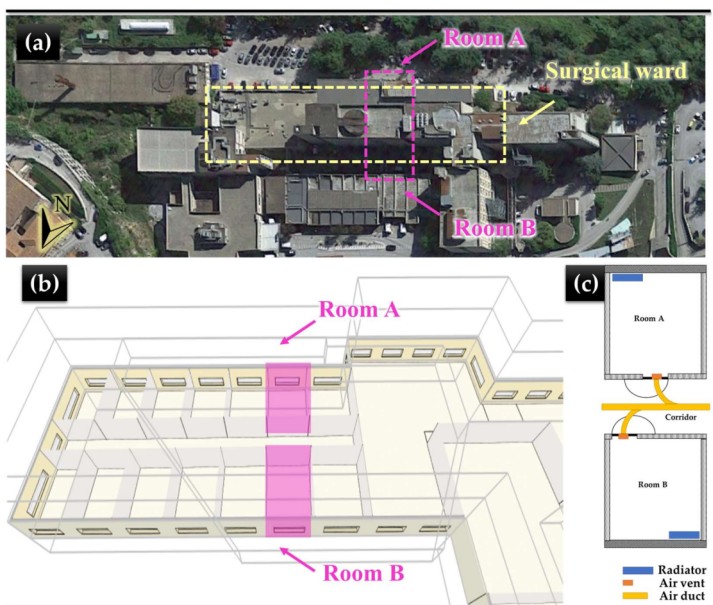

**Figure 3.** View from top of the hospital (**a**); view of analyzed rooms (**b**); HVAC units/ventilation ducts distribution in the rooms (**c**).

**Table 2.** Thermo-physical characteristics of opaque building envelope.

| | $t$ (m) | $\lambda$ (W/mK) | $\rho$ (kg/m$^3$) | $c_p$ (kJ/kgK) | $R$ (m$^2$K/W) | Total $t$ (m) | $U$ (W/m$^2$K) |
|---|---|---|---|---|---|---|---|
| | | | External wall | | | | |
| Cement plaster | 0.01 | 0.72 | 1760 | 840 | | | |
| Brick reinforced | 0.48 | 1.10 | 1920 | 840 | | 0.50 | 1.58 |
| Gypsum plastering | 0.01 | 0.80 | 1300 | 840 | | | |
| | | | Internal wall | | | | |
| Gypsum plastering | 0.01 | 0.80 | 1300 | 840 | | | |
| Hollow brick | | 0.30 | 800 | 1000 | | 0.10 | 1.81 |
| Gypsum plastering | 0.01 | 0.80 | 1300 | 840 | | | |
| | | | Internal floor slab | | | | |
| Flooring | 0.02 | 1.0 | 1200 | 1000 | | | |
| Concrete screed | 0.05 | 1.06 | 2000 | 1000 | | | |
| Concrete slab | 0.05 | 1.6 | 2300 | 1000 | | 0.34 | 1.80 |
| Brick and concrete block | 0.22 | | | | 0.218 | | |
| Lime and gypsum plaster | 0.02 | 0.7 | 1400 | 1000 | | | |

*3.2. Patients' Clothing and Activities*

In order to define the thermal resistance of the clothing, different combinations of clothing have been hypothesized, as compatible as possible to staying in the ward. As shown by the literary review above, the metabolic rate and clothing resistance for hospitalized people could be altered by temperature regulation, vasoconstriction, and vasodilatation. Taking these aspects into account, suitable data were used. In detail, in the cases of bedridden patients, for greater accuracy of the evaluations, the resistance of the whole bed-patient system was considered, as defined by Lin and Deng [33]. The calculation of this thermal resistance included:

- Conventional mattress;
- Cover mattress;
- Cotton sheets;
- Polyester quilt.

Different body coverage rates in bed have been taken into account, as defined by [19], and shown in Figure 4. In addition, long or short pajamas with or without dressing gowns were also considered. Thus, all scenarios defined are reported in Table 3.

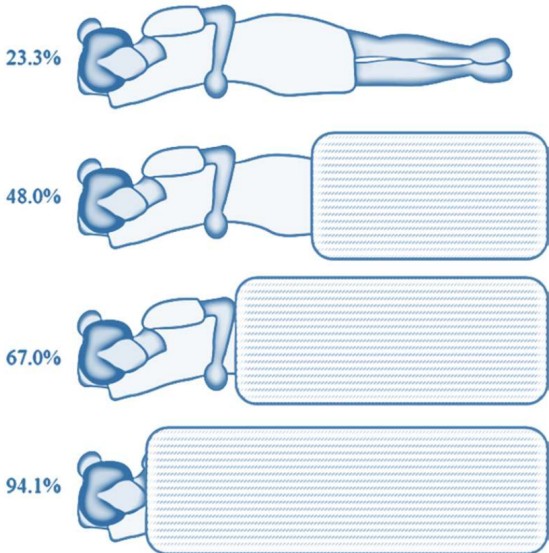

**Figure 4.** Coverage rates for patients in bed.

**Table 3.** Clothing and activity scenarios for patients.

| Scenario | Activity | Clothing | Coverage | Clothing Insulation $I_{cl}$ (clo) | Metabolic Rate M (met) |
|---|---|---|---|---|---|
| 1 | bedridden/asleep | underwear + long pajamas | 94.1% with quilt | 4.56 | 0.7 |
| 2 | bedridden/asleep | underwear + long pajamas | 67.0% with quilt | 2.88 | 0.7 |
| 3 | bedridden/asleep | underwear + long pajamas | 48.0% with quilt | 2.15 | 0.7 |
| 4 | bedridden/asleep | underwear + long pajamas | 23.3% with quilt | 1.57 | 0.7 |
| 5 | bedridden/asleep | underwear + short pajamas | 94.0% with quilt | 4.34 | 0.7 |
| 6 | bedridden/asleep | underwear + short pajamas | 67.0% with quilt | 2.62 | 0.7 |
| 7 | bedridden/asleep | underwear + short pajamas | 48.0% with quilt | 1.65 | 0.7 |
| 8 | bedridden/asleep | underwear + short pajamas | 23.3% with quilt | 1.38 | 0.7 |
| 9 | bedridden/asleep | underwear + long pajamas | 94.1% with bed sheet | 2.56 | 0.7 |
| 10 | bedridden/asleep | underwear + long pajamas | 67.0% with bed sheet | 2.18 | 0.7 |
| 11 | bedridden/asleep | underwear + long pajamas | 48.0% with bed sheet | 1.82 | 0.7 |
| 12 | bedridden/asleep | underwear + long pajamas | 23.3% with bed sheet | 1.57 | 0.7 |
| 13 | bedridden/asleep | underwear + short pajamas | 94.1% with bed sheet | 2.40 | 0.7 |
| 14 | bedridden/asleep | underwear + short pajamas | 67.0% with bed sheet | 1.80 | 0.7 |
| 15 | bedridden/asleep | underwear + short pajamas | 48.0% with bed sheet | 1.43 | 0.7 |
| 16 | bedridden/asleep | underwear + short pajamas | 23.3% with bed sheet | 1.38 | 0.7 |
| 17 | bedridden/awake | underwear + long pajamas | 94.1% with quilt | 4.56 | 0.8 |
| 18 | bedridden/awake | underwear + long pajamas | 67.0% with quilt | 2.88 | 0.8 |
| 19 | bedridden/awake | underwear + long pajamas | 48.0% with quilt | 2.15 | 0.8 |
| 20 | bedridden/awake | underwear + long pajamas | 23.3% with quilt | 1.57 | 0.8 |
| 21 | bedridden/awake | underwear + short pajamas | 94.1% with quilt | 4.34 | 0.8 |
| 22 | bedridden/awake | underwear + short pajamas | 67.0% with quilt | 2.62 | 0.8 |
| 23 | bedridden/awake | underwear + short pajamas | 48.0% with quilt | 1.65 | 0.8 |
| 24 | bedridden/awake | underwear + short pajamas | 23.3% with quilt | 1.38 | 0.8 |
| 25 | bedridden/awake | underwear + long pajamas | 94.1% with bed sheet | 2.56 | 0.8 |
| 26 | bedridden/awake | underwear + long pajamas | 67.0% with bed sheet | 2.18 | 0.8 |

**Table 3.** *Cont.*

| Scenario | Activity | Clothing | Coverage | Clothing Insulation $I_{cl}$ (clo) | Metabolic Rate M (met) |
|---|---|---|---|---|---|
| 27 | bedridden/awake | underwear + long pajamas | 48.0% with bed sheet | 1.82 | 0.8 |
| 28 | bedridden/awake | underwear + long pajamas | 23.3% with bed sheet | 1.57 | 0.8 |
| 29 | bedridden/awake | underwear + short pajamas | 94.1% with bed sheet | 2.40 | 0.8 |
| 30 | bedridden/awake | underwear + short pajamas | 67.0% with bed sheet | 1.80 | 0.8 |
| 31 | bedridden/awake | underwear + short pajamas | 48.0% with bed sheet | 1.43 | 0.8 |
| 32 | bedridden/awake | underwear + short pajamas | 23.3% with bed sheet | 1.38 | 0.8 |
| 33 | bedridden/awake | underwear + long pajamas + dressing gown | - | 1.07 | 0.8 |
| 34 | bedridden/awake | underwear+ short pajamas+ dressing gown | - | 0.92 | 0.8 |
| 35 | sitting/awake | underwear+ short pajamas | - | 0.46 | 1.0 |
| 36 | sitting/awake | underwear+ long pajamas+ dressing gown | - | 1.07 | 1.0 |
| 37 | sitting/awake | underwear+ short pajamas+ dressing gown | - | 0.92 | 1.0 |
| 38 | standing/awake | underwear + long pajamas | - | 0.92 | 1.2 |
| 39 | standing/awake | underwear + short pajamas | - | 0.46 | 1.2 |
| 40 | standing/awake | underwear+ long pajamas+ dressing gown | - | 1.07 | 1.2 |
| 41 | standing/awake | underwear+ short pajamas + dressing gown | - | 0.92 | 1.2 |

The parametric analyses developed with respect to patients' clothing, bedding types, and percentage of body coverage can help in considering the variation of bed-patient system thermal resistance that is usually not a static parameter.

*3.3. Comfort Indices Calculation*

For the evaluation of the static comfort, namely for the calculation of the PMV and PPD, the prescriptions of the ISO 7730 standard [2] were followed. The hourly values of dry-bulb air temperature, $T_a$ (°C), average radiant temperature, $T_r$ (°C), air relative humidity, *R.H.* (%), were carried out from the simulations over one year. These values, combined with clothing insulation $I_{cl}$ (m$^2$ °C/W) and metabolic rate $M$ (W/m$^2$) (Table 3), were defined as follows.

- $T_{cl}$: surface temperature of the dressed body (°C):

$$T_{cl} = 35.7 - 0.0028(M - W) - I_{cl}\{3.96 \times 10^{-8}\left[(T_{cl} + 273)^4 - (T_r + 273)^4\right] + f_{cl}h_c(T_{cl} - T_a)\} \tag{1}$$

- $h_c$: convective heat transfer coefficient (W/m$^2$ °C):

$$\begin{cases} h_c = 2.38|T_{cl} - T_a|^{0.25} & if \ 2.38|T_{cl} - T_a|^{0.25} > 12.1\sqrt{v_{ar}} \\ h_c = 12.1\sqrt{v_{ar}} & if \ 2.38|T_{cl} - T_a|^{0.25} < 12.1\sqrt{v_{ar}} \end{cases} \tag{2}$$

- $f_{cl}$: clothing area coefficient (-):

$$\begin{cases} f_{cl} = 1.00 + 1.29I_{cl} & if \ I_{cl} \le 0.078 \ \text{m}^2°\text{C/W} \\ f_{cl} = 1.05 + 0.645I_{cl} & if \ I_{cl} \ge 0.078 \ \text{m}^2°\text{C/W} \end{cases} \tag{3}$$

- $v_{ar}$: relative air speed (m/s):

$$v_{ar} = v_a + 0.005(M - 58.15) \tag{4}$$

- $p_a$: partial pressure of the air vapor (Pa):

$$p_a = 10 \ R.H. \ e^{\frac{(16.6536 - 4030.183)}{(T_a + 235)}} \tag{5}$$

where $W$ is the external work (W/m$^2$) that is assumed equal to zero for the activities considered.

Once these quantities are known, it is possible to calculate the *PMV* (-) and the *PDD* (%) through Equations (6) and (7).

$$
\begin{aligned}
PMV = \ & (0.303e^{-0.036M} + 0.028) \times \{(M-W) - 3.05 \times 10^{-3}[5733 - 6.99(M-W) - p_a] - 0.42[(M-W) - 58.15] \\
& - 1.7 \times 10^{-5}M(5867 - p_a) - 0.0014M(34 - T_a) - 3.96 \times 10^{-8}f_{cl}\left[(T_{cl} + 273)^4 - (T_r + 273)^4\right] \\
& - f_{cl}h_c(T_{cl} - T_a)\}
\end{aligned}
\tag{6}
$$

$$
PDD = 100 - 95e^{-(0.03353PMV^4 - 0.2179PMV^2)}
\tag{7}
$$

Similarly, for the assessment of adaptive comfort, the prescriptions of the ASHRAE-55 standard [3] were followed. In particular, the hourly value of the operative temperature $T_o$ (°C) for the rooms under examination, over the whole observation period, was determined from the simulations. In order to define the lower ($T_{low\ limit}$) and upper ($T_{up\ limit}$) limit values (°C) of the aforesaid temperature, Equations (8) and (9) were used. This is because the hospital could be considered a fully mechanically-controlled (FMC) building, so the equations proposed by the standard for correlating the outdoor air temperature and indoor limit values were not suitable. The proper model, developed for "HVAC buildings", was used, as shown in [34]. These temperature limits define the area where <10% of people were dissatisfied with the microclimate.

$$
T_{up\ limit} = 0.11T_{pma} + 23.95
\tag{8}
$$

$$
T_{low\ limit} = 0.11T_{pma} + 18.95
\tag{9}
$$

where $T_{pma}$ is the average value of the outdoor air temperature of the 30 days preceding the one in question.

*3.4. Analyzed Periods and Boundary Conditions*

The simulations were carried out by using the hourly climate file of Isernia.

The weather file showed a minimum outdoor dry-bulb temperature of −5.9 °C (on 12 January) and a maximum of 35.3 °C (on 19 August). The average monthly outdoor air temperature is shown in Figure 5a. The average temperature was above 18.0 °C during the hottest months (June–September) and ranged from 5 °C to 15 °C in the colder months. The relative humidity was greater than 70% for 75% of the hours over the year, with an average annual value of 80%. The prevalent direction of the wind was south-north, and the maximum wind speed was greater than 8.0 m/s and occurred in March.

In order to analyze the results, the season periods were organized as follows: summer period from 15 June to 15 September; spring from 16 April to 14 June; autumn period from 16 September to 31 October; and winter from 1 November to 15 April. Among them, four days were chosen as a reference for their season, since they appeared to be representative of the period:

- 10 February, reference in winter;
- 15 May, reference in spring;
- 15 July, reference in summer;
- 10 October, reference day of autumn period.

These reference days were chosen after analyzing all available days, taking into consideration the days with outdoor parameter trends similar to the median trend of the season. Moreover, the chosen references are similar to those used in another paper [35]; furthermore, the approach for showing results is similar to that carried out in an analogous climate study.

The hourly trend of the main climatic parameters for the aforementioned days are depicted in Figure 5b (outdoor dry-bulb temperature), Figure 5c (relative humidity), and Figure 5d (global solar radiation on horizontal plane).

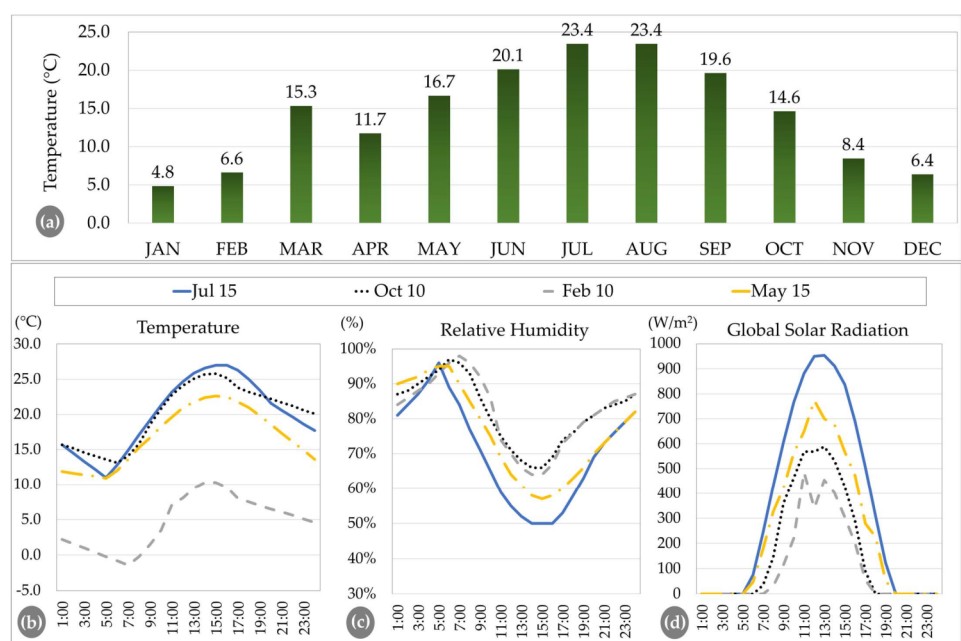

**Figure 5.** Average monthly outdoor air temperature (**a**); average hourly outdoor air temperature (**b**); relative humidity (**c**); and global solar radiation on horizontal surface (**d**).

Considering the methodological approach applied to the case study, the results could be representative of most southern Italian hospitals built in the late seventies since the climate boundary conditions and the HVAC-envelope system are similar. For instance, the hospital in Naples [12] is characterized by similar stratigraphy and thermal transmittance values (e.g., the external walls are made of hollow bricks and air gaps with an overall thermal transmittance of 1.3 W/m²K) with various HVAC typology systems for different served areas, as in the case study of the present paper. Similar characteristics in terms of the whole building could be found in other hospitals constructed in the same period in Europe [36]. However, it should be stressed that there are very few datasets available for the climatic zone regarding the hospitals building typology, as declared by Carnieletto et al. in a recent paper of 2021 [37]. On the other hand, the limitation on the results regards first, the type of patients; indeed, they are bedridden partially healthy patients in the surgery department, but it does not refer to very sensitive or fragile patients with special requirements like handicapped, sick, very young children, or elderly persons. Moreover, the thermal comfort was evaluated by means of two different accredited models [2,3] without considering the localized discomfort.

## 4. Results

In this section, the results obtained from the application of the comfort model, traditional or adaptive, are proposed for the rooms under investigation.

### 4.1. Daily Analysis

The scenarios shown in Table 3 differ by level of activity (or metabolic rate); therefore, they are not suitable for all hours of the day. For example, scenarios with a patient out of bed are unlikely during night hours. A further difference among the scenarios concerns the type of clothing adopted by the patient; this shows that not all scenarios are appropriate for all four days (or season) analyzed, i.e., a patient out of bed with only short pajamas in the winter period is implausible. Table 4 shows which periods of the day and which representative days the various scenarios were considered. By way of example, Scenario 1 considered the typical winter and autumn day during night hours, from 7:00 pm to 6:00 am; meanwhile, Scenarios 2 to 8 evaluated only the autumn period during nighttime. Again, on the basis of the data reported in Table 4, Scenario 39 is representative of a condition that

is good in the summer and autumn periods (and so in corresponding representative days) during the daytime.

**Table 4.** Summary of daily calculation scenarios.

| Day | Day | | Night | |
|---|---|---|---|---|
| | Scenario | Time Slot | Scenario | Time Slot |
| 15 May | 25–32 36–38; 40–41 | 7:00–18:00 9:00–20:00 | 9–16 | 19:00–6:00 |
| 15 July | 25–32 35–41 | 7:00–18.00 9:00–20:00 | 9–16 | 19:00–6:00 |
| 10 October | 17–24 35–41 | 7:00–18:00 9:00–20:00 | 1–8 | 19:00–6:00 |
| 10 February | 17–20 36; 40 | 7:00–18:00 9:00–20:00 | 1 | 19:00–6:00 |

For the Fanger model, the hourly values of PMV are shown for all configurations and both rooms by means of radar charts and histograms. For a better understanding, we always highlight the comfort zone (green areas). For the adaptive model, the hourly trend of operative temperature is depicted, for both rooms, always highlighting the limit values (red lines).

4.1.1. Spring Day (15 May)

First of all, the conditions of the bedridden patient, asleep or awake, were analyzed for all hours of the day. Figure 6 reports the case of a patient with long or short pajamas and with different coverage rates. On the radial axis, the PMV is shown; on the circular axis, the hours are shown. The latter refers to ante meridiem (AM) or post meridiem (PM), depending on the label. The comfort zone, $-0.5 < \text{PMV} < 0.5$, is depicted in green. The scenario considered for a given hour of the day refers to Table 4. For instance, considering the label "AM–coverage 23.3%" for the long pajamas case refers from 12:00 to 6:00 a.m. in Scenario 12 (bedridden/asleep) and from 7:00 to 11:00 a.m. in Scenario 28 (bedridden/awake). Analyzing the long pajamas case, only the 94.1% coverage from 12:00 to 6:00 p.m. showed a PMV > +0.5. On the other hand, in the same hours, all coverage percentages ≤ 67.0% resulted in a PMV ≤ −0.5. From 7:00 to 11:00 p.m., only the 23.3% coverage showed a PMV ranging from −0.6 to −0.9. The short pajamas case highlights more scenarios below the lower comfort limit, while only the 94.1% bed sheet coverage rate from 12:00 to 6:00 p.m. achieved a PMV greater than the upper limit (+0.5). In general, the PMV values passing from room B (north-facing) to A (south-facing) increased ~+0.1.

In Figure 7, the PMV histograms are shown together with operative temperature ($T_o$) and relative humidity (RH) trends for the day and nighttime. During the nighttime, the mean values of micro-climate parameters were:

- For room B, $T_o \approx 22.0$ °C and RH $\approx 62\%$;
- For room A, $T_o \approx 22.4$ °C and RH $\approx 61\%$.

Until 23:00, $T_o$ was constant, around 23 °C, and then decreased by ~1 °C, and remained constant until 6:00. After the decrease of $T_o$, the cold sensation increased for both long (9–12) and short pajama (13–16) scenarios. For the PMV, the further away from the comfort zone, the lower the coverage levels were. In order to stay in the comfort zone all night long, the short or long pajamas had to be combined with the maximum coverage percentage of the bedsheet. The aforementioned reduction of $T_o$ determined that the long pajama with 67.0% coverage was not suitable.

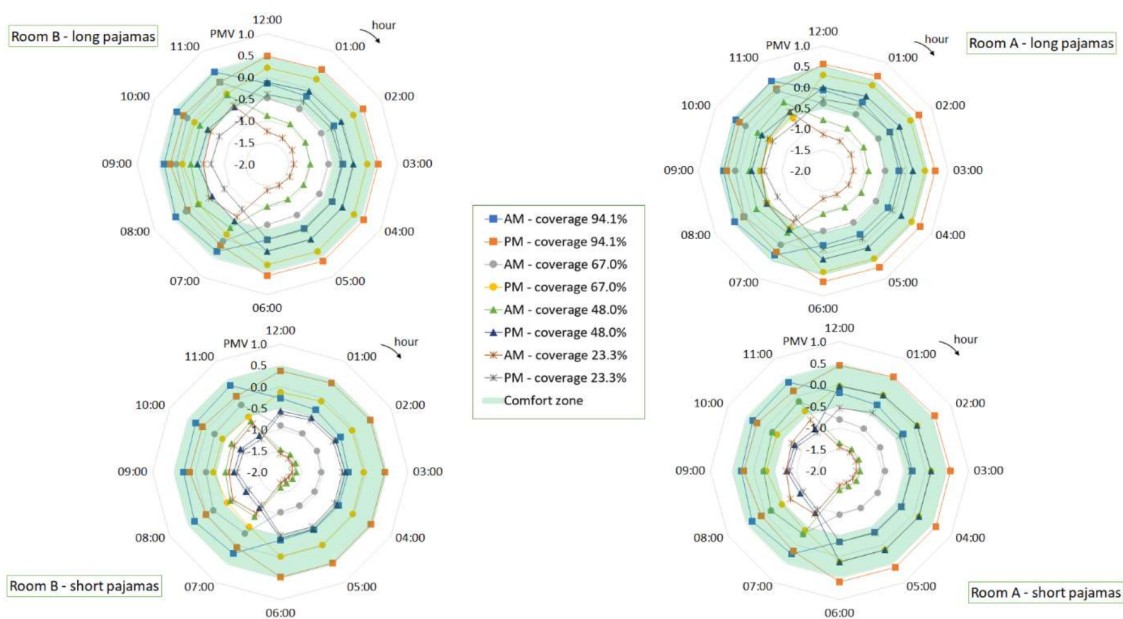

**Figure 6.** Hourly PMV of bedridden patients: different coverage rates and clothing on the spring day.

Regarding the daytime charts, the mean $T_o$ was 23.0 °C in room A and 22.6 °C in room B, with an almost constant trend. The mean RH was 61% and 62% for room A and room B, respectively. During the day, the conditions were more favorable with respect to the nighttime, both for the long pajamas scenarios (25–28) and the short pajamas scenarios (29–32). Considering the long pajamas scenarios in room B, the thermo-hygrometric conditions were not acceptable for a coverage of 94.1% (scenario 25) when the $T_o$ was $\geq$ 23.0 °C, from 14:00 to 18:00; and when the coverage was 23.3% with a $T_o$ < 22 °C, at 7:00. In the short pajamas scenarios, in room B, when $T_o$ was < 23.0 °C (from 7:00 to 15:00) and there was a lower coverage percentage of bedsheet, the PMV was < −0.5. In room A, the cool feeling discomfort (from 7:00 to 12:00 in scenarios 31 and 32) and the warm feeling discomfort (from 12:00 to 18:00 in scenarios 25 and 29) were more pronounced with respect to room B.

Finally, the results for the out-of-bed patient during the daytime are shown in Figure 8. In both rooms A and B, the only discomfort condition ("slightly cool") was given by Scenario 37, in which the patient was sitting and wearing short pajamas in combination with a dressing gown.

For the same typical spring day, an analysis of the adaptive model was conducted. With the hourly data of the weather file used for the simulations, the daily outdoor air temperatures were calculated, and the $T_{pma}$ was equal to 12.3 °C. Figure 9 shows the results. The upper and lower limits were 25.3 °C and 20.3 °C, respectively. It can be noted that the inner $T_o$ during 24 h was always between the upper and lower limit both for room A and B. $T_o$ varied in the range 21.4–23.7 °C for room A and 21.1–23.3 °C for room B. Contrary to the static comfort model, in this case for all hours of the day, the patient was in comfort conditions, but the adaptive model did not consider the patient's hospitalization conditions in any way.

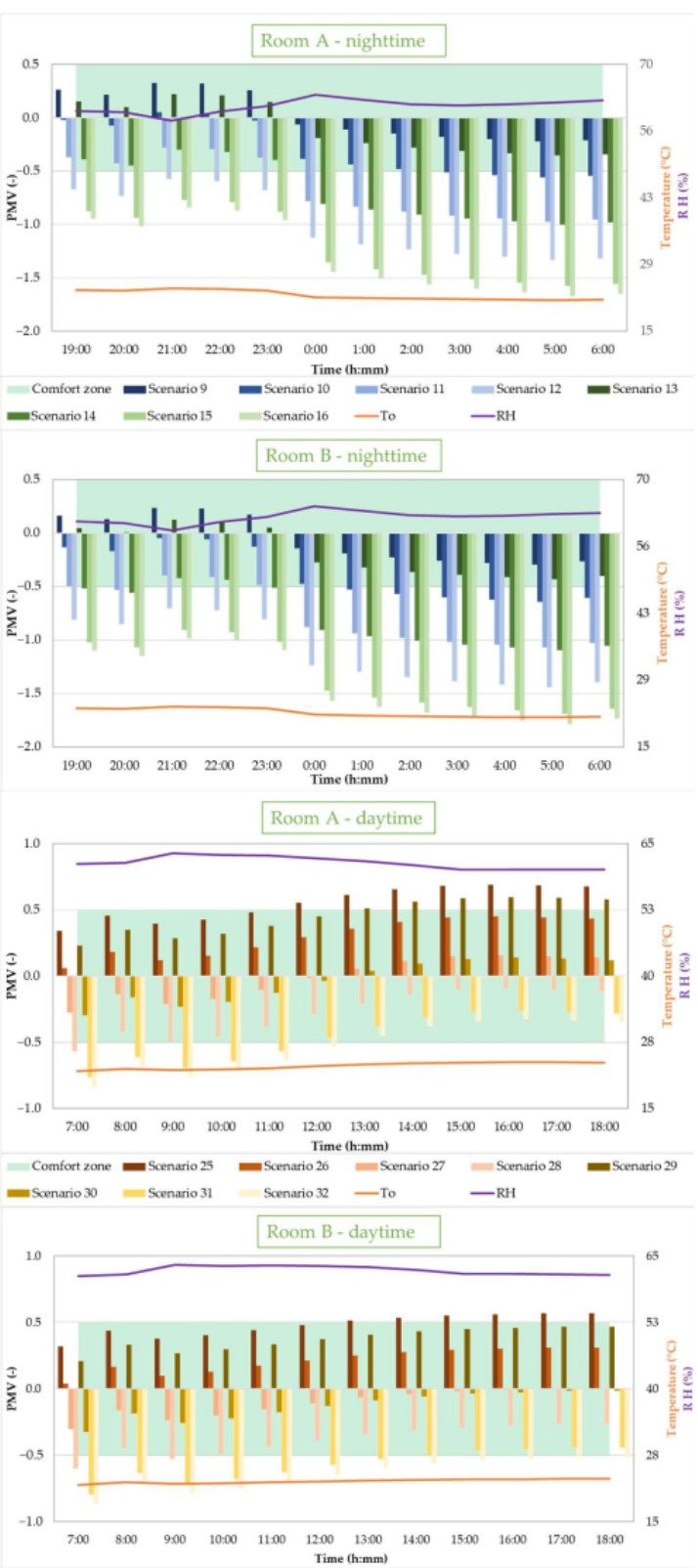

**Figure 7.** Hourly operative temperature, relative humidity, and PMV in rooms A and B with a bedridden patient: different scenarios in the spring day.

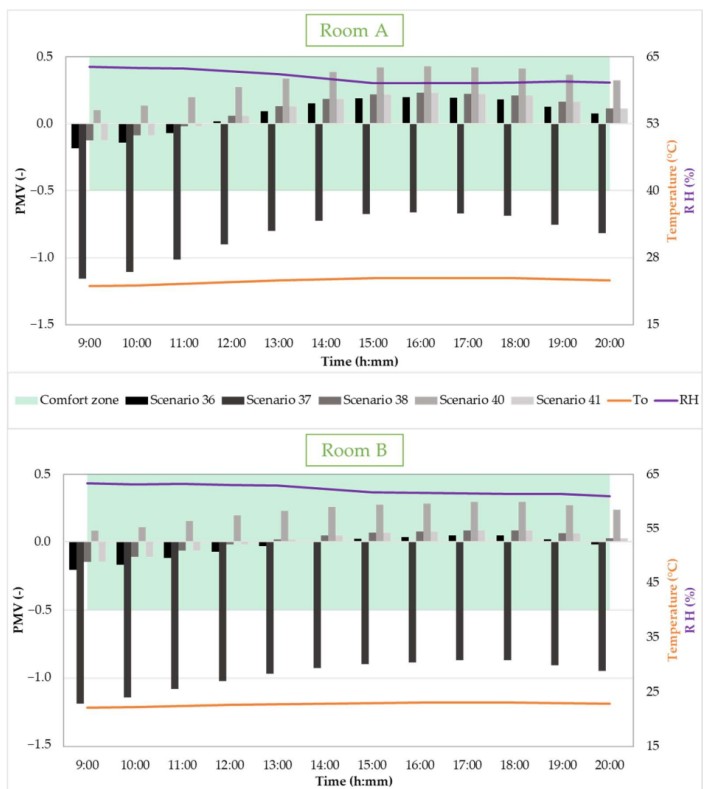

**Figure 8.** Hourly operative temperature, relative humidity, and PMV in rooms A and B with an out-of-bed patient: different scenarios in the spring day.

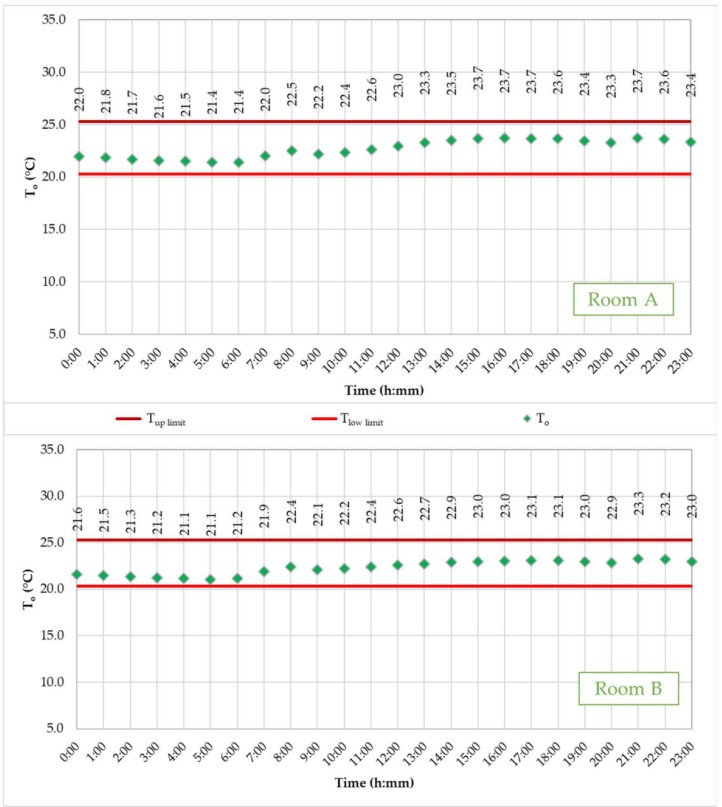

**Figure 9.** Hourly operative temperature for the spring day.

### 4.1.2. Summer Day (15 July)

The reference summer day in Figure 10 shows the hourly PMV of the bedridden patient, asleep or awake, for the scenarios with long and short pajamas. For almost all scenarios with long pajamas (in room A or B), there was a PMV greater than +0.5: the discomfort was expressed as "slightly warm" or "warm" environment, according to the Fanger scale. Only in the few hours with 23.3% of coverage was there a PMV lower than 0.5, from 2:00 a.m. to 6:00 a.m. and from 7:00 p.m. to 8:00 p.m. Regarding the patient with short pajamas, the comfortable condition was reached only with the 23.3% or 48.0 % coverage during the night hours.

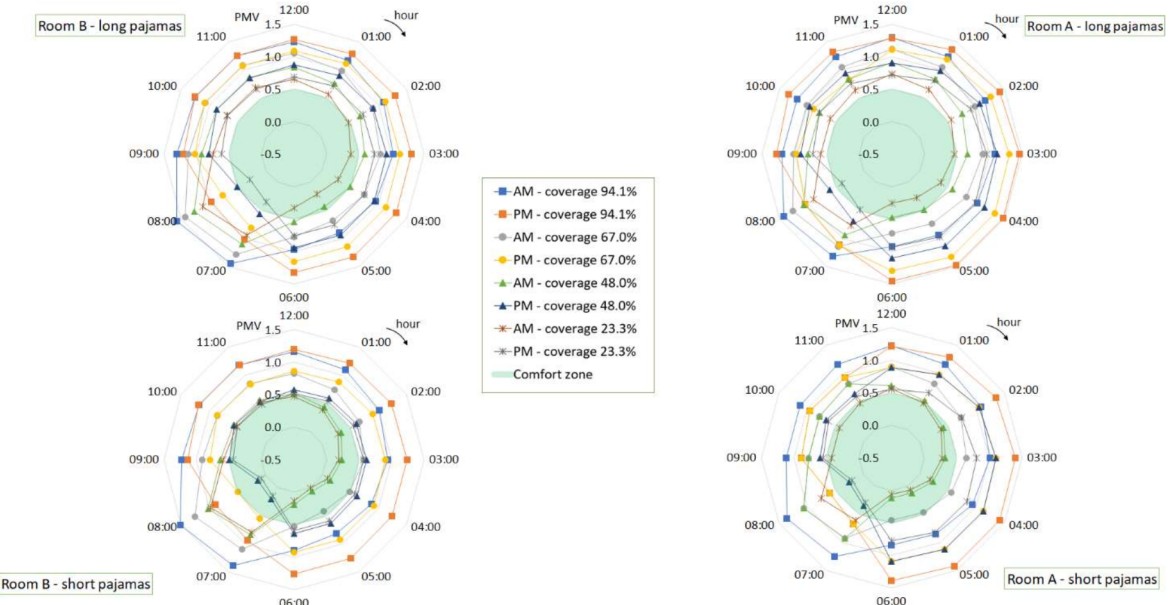

**Figure 10.** Hourly PMV of bedridden patient: different coverage rate and clothing in the summer day.

In more detail, in Figure 11, it can be seen that during the night, $T_o$, starting from 26.4 °C in room B and 26.8 °C in room A, underwent a slight increase (about +1.0 °C) after 20:00, when the air conditioning system was switched off, and then decreased again after midnight. The mean value of $T_o$ was 27.0 °C in room B and 27.2 °C in room A. Furthermore, RH had a tendency to decrease over the observation period, passing from 61–62% to 52–53%. The increase in $T_o$ resulted in a corresponding increase in the PMV. In the scenarios with coverage percentages of 94.1% and 67.0% for both long and short pajama cases (Scenario 9–10 and 13–14), the PMV was higher than +0.5. Similarly, the scenarios with a lower coverage rate, when paired with long pajamas (Scenario 11 and 12), showed a PPD greater than 10%. Meanwhile, combinations with a low percentage of coverage in the case of short pajamas (Scenario 15 and 16) were within the comfort zone, for all night hours in room B and for the periods 19:00–20:00 and 1:00–6:00 in room A. Considering the midnight time, passing from room A (warmer) to B (colder), the PMV values decreased (nearing thermal neutrality) approximately:

- −5% in Scenario 9 and −5% in Scenario 13;
- −6% in Scenario 10 and −9% in Scenario 14;
- −8 in Scenario 11 and −15% in Scenario 15;
- −12% in Scenario 12 and −16% in Scenario 16.

The most marked percentage variations occurred in the cases with low coverage rates. This trend could also be seen in the other night hours. The patient with low coverage percentage found comfort conditions easier when the microclimate parameters changed in summer.

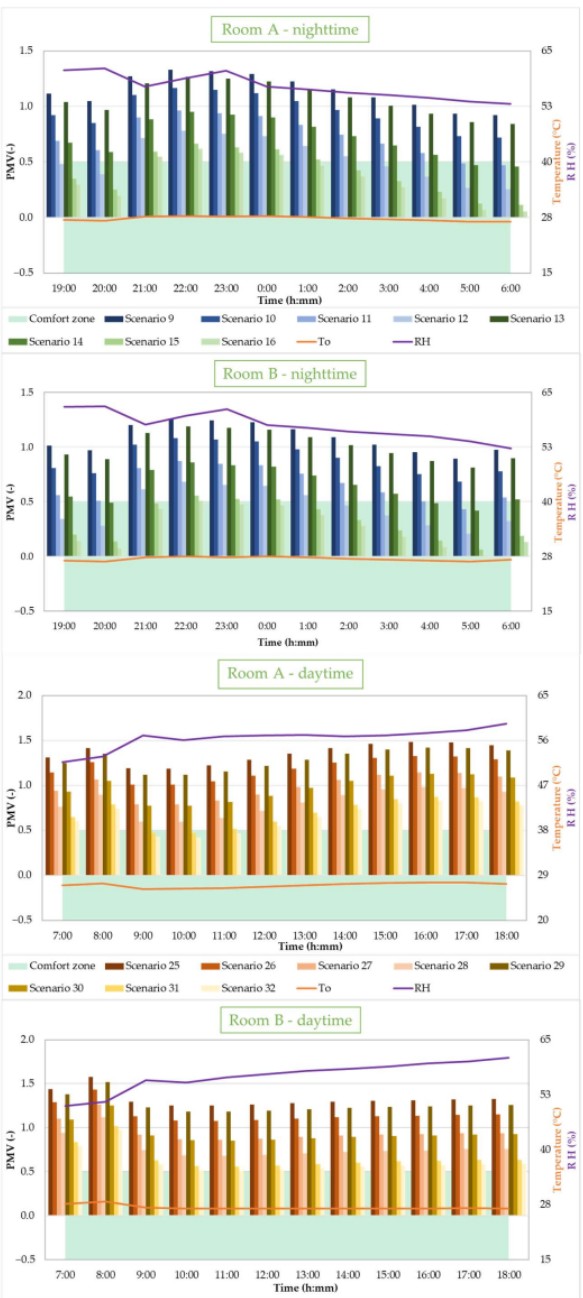

**Figure 11.** Hourly operative temperature, relative humidity, and PMV in rooms A and B with a bedridden patient: different scenarios in the summer day.

Regarding the daytime long pajama cases (Scenarios 25–28), despite the fact that after 8:00 in the morning, the HVAC systems are on, with $T_o \approx 26.5$ °C and $T_a \approx 25.1$ °C, the sensation was always between "slightly warm" to "warm", in both rooms. This highlights the inadequacy of the long pajamas for indoor microclimatic conditions in summer. The situation was improved in the case of short pajamas (scenarios 29–32), but the PMV was greater than +0.5 for each level of coverage. The increase of RH during the morning (from 52% to 59% in room A and from 51% to 61% in room B) did not play a positive role with respect to these judgments.

In cases where the patient was out of bed (Figure 12), the application of the Fanger model suggested that positive judgments regarding the microclimate occurred where the person wore only short pajamas, both in standing and sitting situations (Scenarios 35, 37, and 39). Moreover, in this case, the PMV values in room A were greater than the ones in

room B starting from midday: the difference was most marked when the body was more uncovered. For instance, at 15:00, the percentage difference between the PMV in rooms A and B were:

- −14% in Scenario 40;
- −17% in Scenarios 36, 38, and 41;
- −44% in Scenario 37;
- −51% in Scenario 39;
- −94% in Scenario 35.

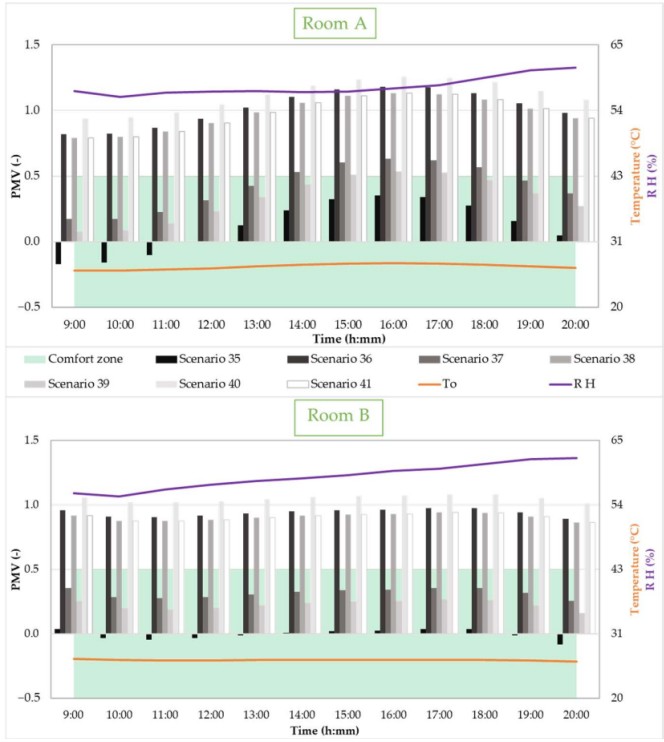

**Figure 12.** Hourly operative temperature, relative humidity, and PMV in rooms A and B with an out-of-bed patient: different scenarios in the summer day.

Analyzing the application of the adaptive comfort model on 15 July (Figure 13) confirmed that the $T_o$ is equal or greater than the upper limit for most hours of the day in both rooms A and B. This demonstrates that the situation of discomfort persisted. For the day taken into account, $T_{low\ limit}$ = 21.4 °C and $T_{up\ limit}$ = 26.4 °C. Despite room B reaching a peak of $T_o$ (28.2 °C) at 8:00, the sum of the hourly differences between indoor $T_o$ and its upper limit, for the whole day, was 13.6 °C and 18.5 °C in room B and room A, respectively. This shows that the room in which there are greater discomfort conditions is the one facing south (room A).

### 4.1.3. Autumn Day (10 October)

Applying the Fanger model for the typical autumn day, the obtained results are shown in Figures 14–16.

In Figure 14, showing a bedridden patient, the coverage level of 94.1% was in combination with long pajamas only. It is evident that, in both rooms, with both the short and long pajamas, there was a situation of discomfort tending to "hot". In room B, with a long pajama, the "neutral" condition could be reached with 23.3% in the morning hours; with a short pajama, the "neutral" condition could be reached with 23.3% and 48.0%, during the same hours. On the other hand, in room A, comfort conditions were hardly ever reached, both with long and short pajamas, for any coverage rate.

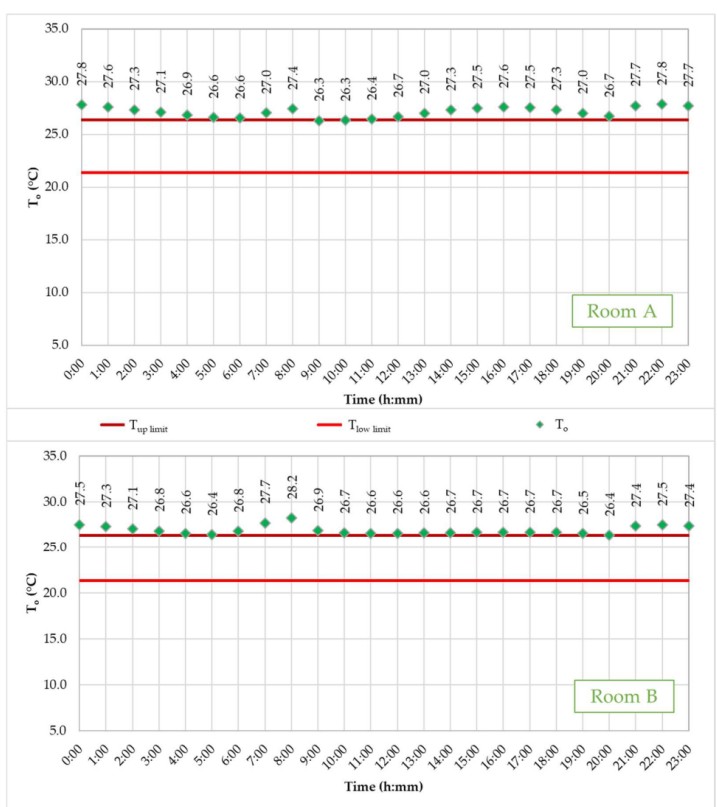

**Figure 13.** Hourly operative temperature for the summer day.

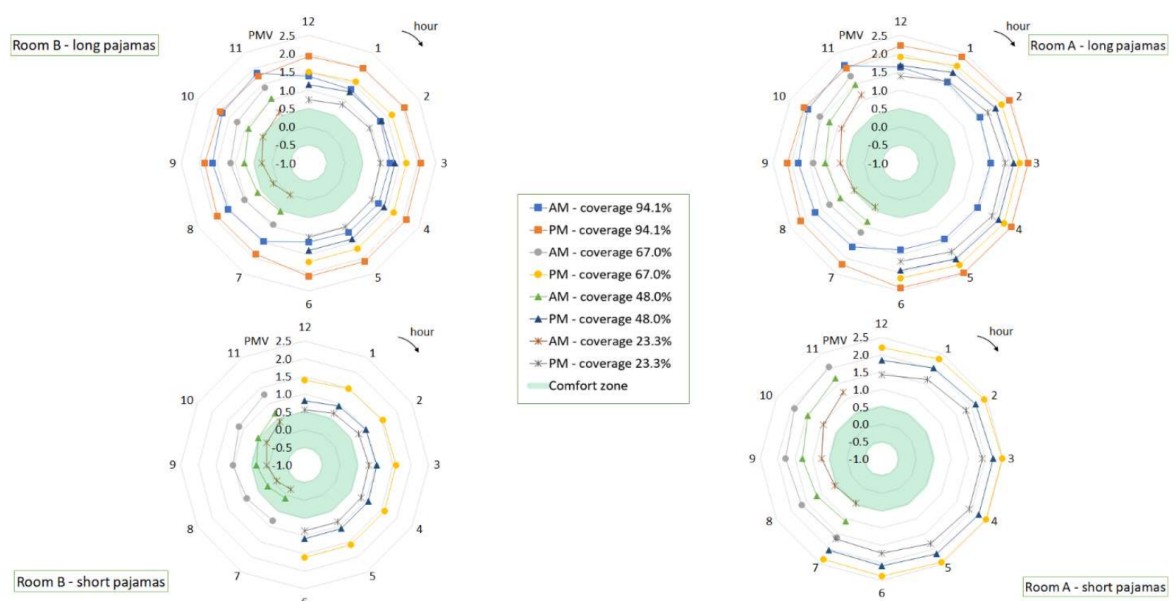

**Figure 14.** Hourly PMV of bedridden patient: different coverage rate and clothing in the autumn day.

Considering the trend of $T_o$ and RH in Figure 15, it can be seen that during the night, in the early hours (19:00–23:00), $T_o$ was around 27.0 °C in room B and 29.0 °C in room A, with an RH ranging 75–77% in room B and 68–72% in room A. This implies that any combination of clothing and cover in the early hours of the night is highly unacceptable, resulting in a perception going from "slightly warm" to "warm", especially in south exposed rooms (room A). After 23:00, there was a progressive decrease in RH and $T_o$; therefore, the PMV decreased, but the thermal discomfort sensation persisted if the level of coverage was 67.0%

or more, in room A, and 94.1% in room B, with short or long pajamas. At the same time, the situation of maximum uncovering with short pajamas (Scenario 8 in room B) became critical in the opposite direction: "slightly cool" sensation. Scenarios 7 and 8, over the 12-hour nighttime of the autumn day, showed a great variation of thermal sensation, from "slightly warm" to "slightly cool".

Analyzing the daytime results for the bedridden patient (Figure 15), the RH ranged 63–73% and 60–67%, in room B and A, respectively, while the $T_o$ varied between 24 °C and 27 °C in room B and between 25 °C and 30 °C in room A. In this time period, only in the early hours with minimum coverage rates (Scenarios 20 and 23) was there a situation of comfort, both with long and short pajamas. In all other cases, there was always a PMV greater than +0.5 and, therefore, a "slightly warm" to "warm" sensation. This trend was linked to the fact that the $T_o$ and RH progressively increased during the day.

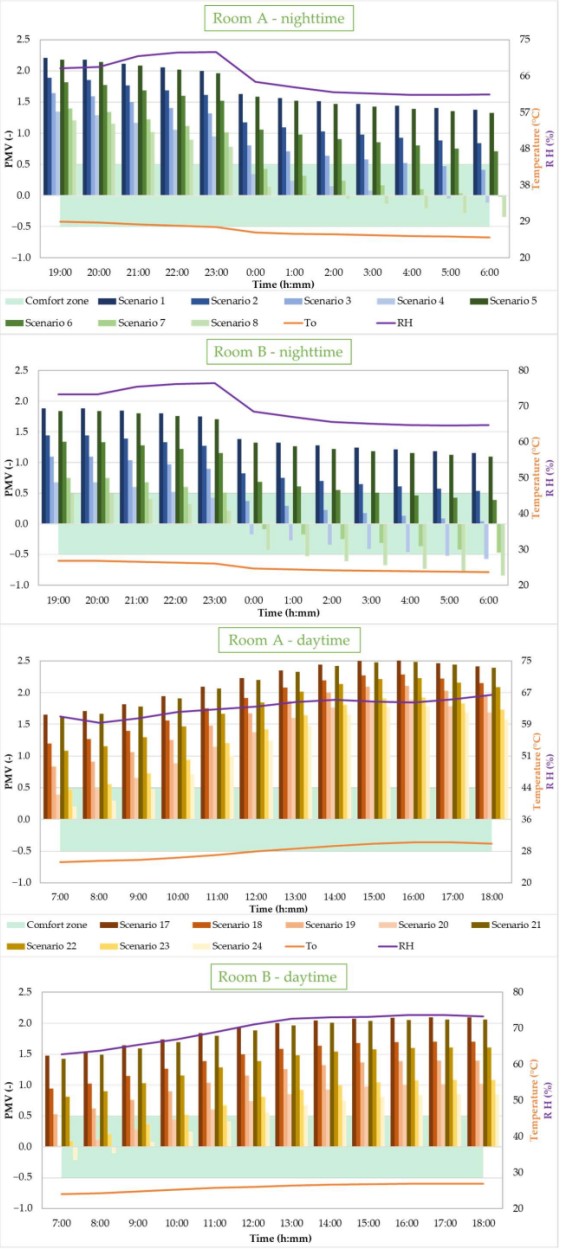

**Figure 15.** Hourly operative temperature, relative humidity, and PMV in rooms A and B with a bedridden patient: different scenarios in the autumn day.

In the case describing the patient out of bed in room A, for all scenarios and most hours, PMV values were > +0.5 (Figure 16); only the combination involving short pajamas (Scenarios 35, 37, and 29) until 10:00 were within the comfortable range. In room B, the PMV values were reduced by 1–1.5 compared to room A. Over the day, Scenario 35 (sitting patient with short pajama) showed a PMV going from −0.6 to +0.3.

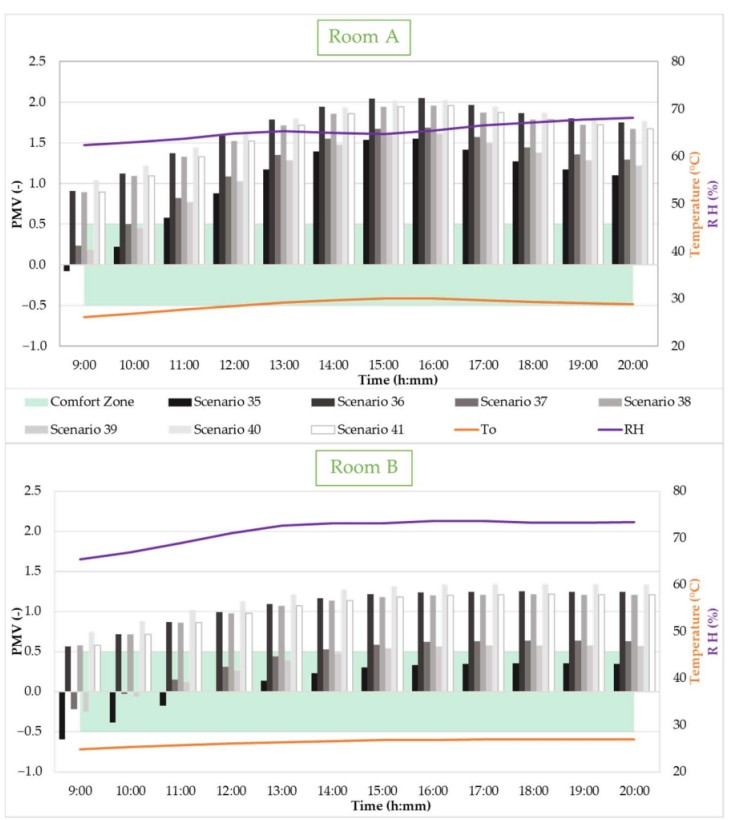

**Figure 16.** Hourly operative temperature, relative humidity, and PMV in rooms A and B with an out-of-bed patient: different scenarios in the autumn day.

Regarding the adaptive model (Figure 17) for the autumn day, the $T_{low\ limit}$ was equal to 20.9 °C, and $T_{up\ limit}$ was equal to 25.9 °C. The mean value of $T_o$ over the whole day was 25.5 °C in room B and 27.6 °C in room A. Until midday in both rooms, the conditions were inside or on the upper limit. In the other hours of the day, $T_o$ values were greater than the limit value. This behavior was more marked in room A. In room B, for 12 h, the $T_o$ was greater than $T_{up\ limit}$, with a sum of differences of 8.7 °C. In room A, there were 18 h in which $T_o$ was greater than $T_{up\ limit}$, with a sum of differences of 42.7 °C.

The results show a strong difference in thermos-hygrometric conditions between the two analyzed rooms.

### 4.1.4. Winter Day (10 February)

For the reference winter day, the results of the application of the Fanger model are shown in Figures 18–20.

In order to evaluate the PMV in the nighttime slot, only the scenarios with long pajamas were taken into account since, in this period, it is rather unlikely to wear short pajamas. In Figure 18, it can be seen that the 94.1% coverage rate fell within the satisfaction range from 7:00 to 11:00 pm. For the daytime slot, on the other hand, the 94.1% coverage rate determined a feeling closely to "slightly warm" in both rooms, while 67.0% was the optimal percentage of coverage in both rooms, in addition to 48.0% in room A.

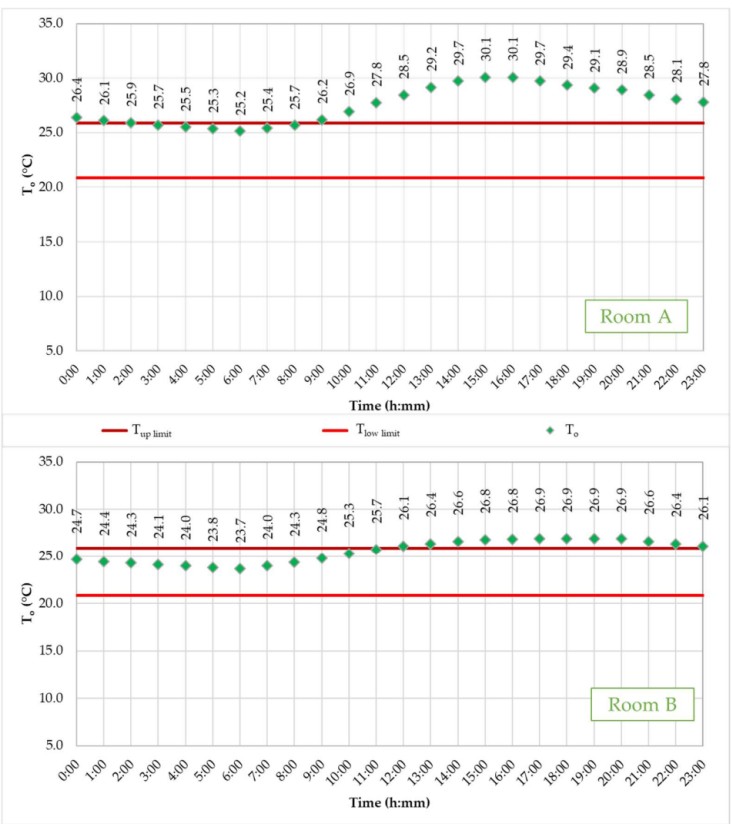

**Figure 17.** Hourly operative temperature for the autumn day.

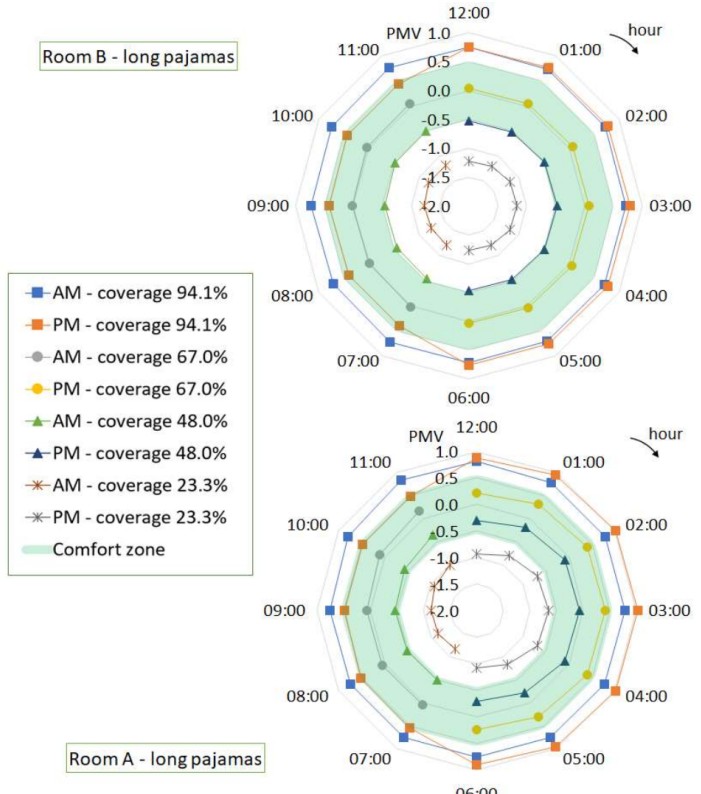

**Figure 18.** Hourly PMV of bedridden patient: different coverage rate in the winter day.

In more detail, by analyzing Figure 19, it can be seen that during the night, with a constant $T_o$ around 20 °C and 21 °C in room B and room A, respectively, and an RH decreasing from 45% to 35% approximately, a neutral thermal sensation was guaranteed. Indeed, in room A, the PMV values were from +0.5 to +0.4, while in room B, they were +0.4 on average.

During the daytime, the $T_o$ was constant at the same night values, with a small increase around 14:00 (+1 °C) in room A, while the RH from the 35% nighttime value progressively increased up to approximately 45%. In both rooms, the most critical conditions, "slightly warm" and "slightly cool", were achieved by using the maximum and minimum coverage rate (Scenario 17 and Scenario 20, respectively) when the patient was in bed. Meanwhile, 48% and 67% coverage rates turned out to be optimal for the thermo-hygrometric comfort of the bedridden patient in the daytime.

Regarding the out-of-bed conditions (Figure 20), Scenarios 36 (sitting) and 40 (standing) demonstrated that the indoor microclimate could be evaluated as "slightly cool" all day long by the seated patient in room B.

In this case, the adaptive model could not be applied since the $T_{pma}$ was lower than 10.3 °C; therefore, it was outside the range of applicability.

*4.2. Seasonal Results*

Following the line defined to show the previous daily results, in these subsections, the results are presented on a seasonal basis. Regarding the assessment of the PMV, all calculated hourly values for the whole season period and for each type of scenario were taken into account. Table 5 summarizes the scenarios used and the relative periods. Since there are a large number of data, for a better presentation, statistical box-plot graphs were developed for the statistic parameters, always highlighting the comfort zone in green. For the adaptive analysis, the operative temperature with different colors in different seasons is plotted, always fixing the limits values as red lines.

**Table 5.** Summary of seasonal calculation scenarios.

| Season | Reference Period | Scenario | |
| --- | --- | --- | --- |
| | | Night (19:00–6:00) | Day (7:00–18:00) |
| Spring | 16 April–14 June | 9–16 | 25–32; 36–38; 40–41 |
| Summer | 15 June–15 September | 9–16 | 25–32; 36–41 |
| Autumn | 16 September–31 October | 1–8 | 17–24; 35–41 |
| Winter | 1 November–15 April | 1–4 | 17–20; 36; 40 |

In order to better manage and represent the high number of data, the results of the PMV are shown with a statistical approach by means of "box and whiskers plots". In this representation, the "x" identifies the mean value of the distribution; the line inside the box, the median; the lower limit of the box, the first quartile ($Q_1$); and the upper limit, the third quartile ($Q_3$). Finally, the two whiskers are equal to the minimum and the maximum values of the distribution, excluding any outliers.

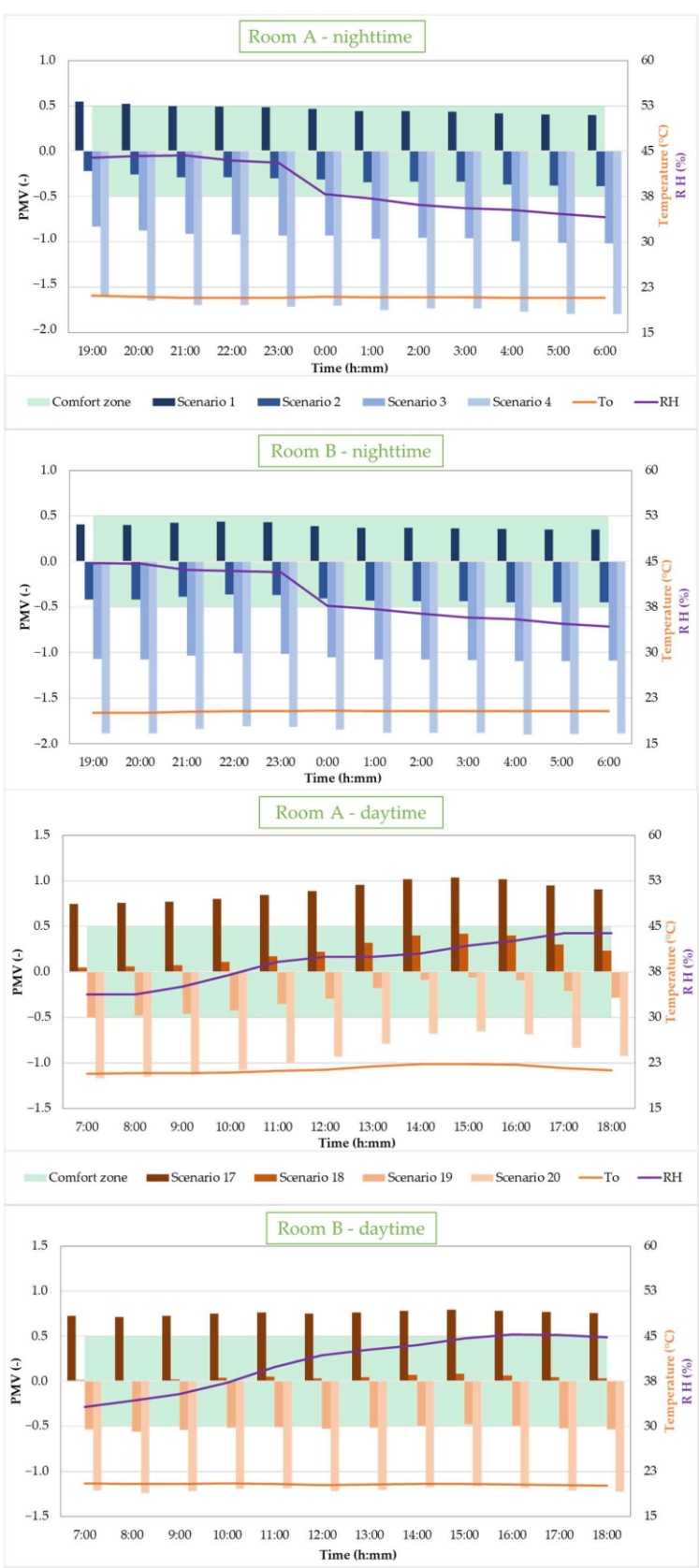

**Figure 19.** Hourly operative temperature, relative humidity, and PMV in rooms A and B with a bedridden patient: different scenarios in the winter day.

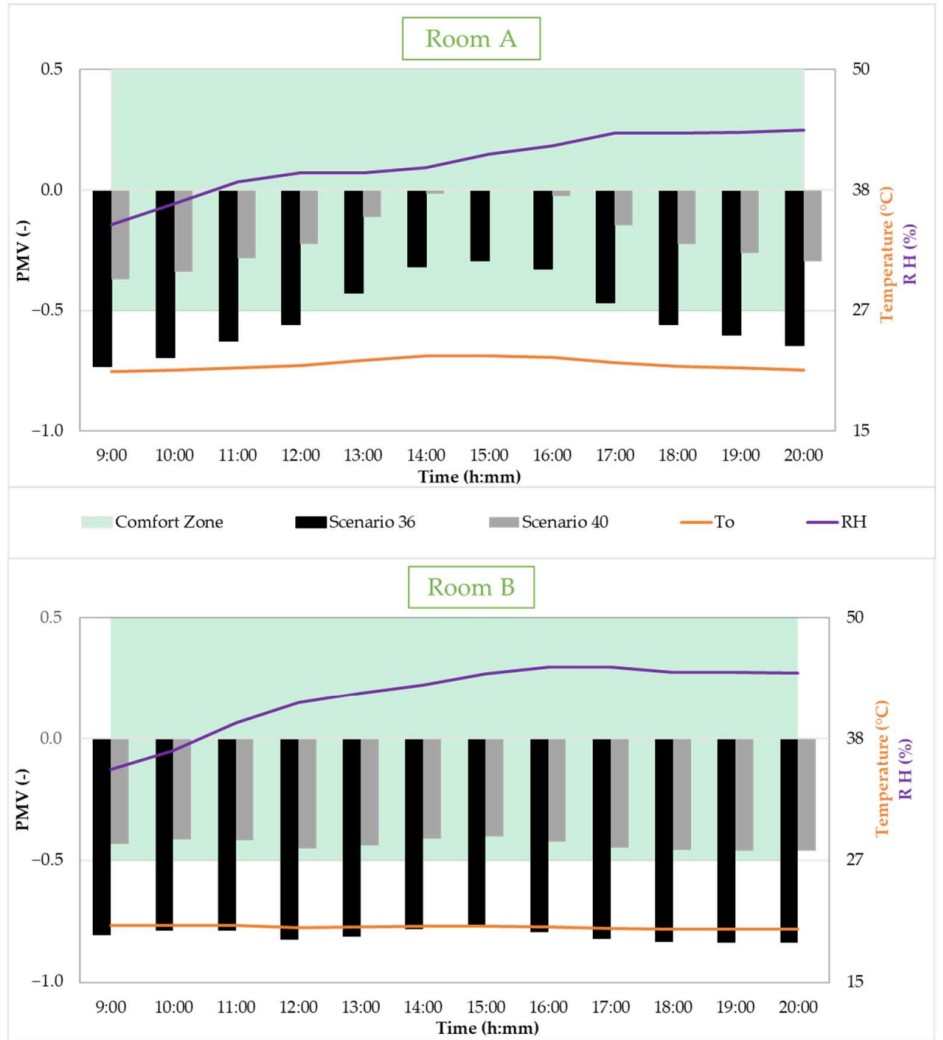

**Figure 20.** Hourly operative temperature, relative humidity, and PMV in rooms A and B with an out-of-bed patient: different scenarios in the winter day.

### 4.2.1. Spring (16 April–14 June)

Figure 21 depicts the box and whiskers graphs for rooms A and B for all scenarios analyzed. In particular, Scenarios 9–16 refer to night hours (19:00–6:00) with a bedridden patient, while Scenarios 25–32; 36–38; 40–41 refer to daily hours (7:00–18:00) with the patient bedridden or out of bed. The statistic parameters depicted in Figure 21 were evaluated on the basis of the hourly values assumed by the independent variables during the entire spring period and taking into account the limits in the use of the scenarios during each day, as reported above.

In the case of scenarios with long pajamas during the nighttime (Scenarios 9–12), the values of the mean PMV for the coverage rate (94.1% to 67.0%) were inside the comfort range (±0.5), in room B. Meanwhile, for room A, this was true for 48.0% of the coverage rate. The interquartile range (IQR) increased with the decrease of coverage rate:

- 0.9 in room B and 0.7 in room A for Scenario 9;
- 0.9 in room B and 0.9 in room A for Scenario 10;
- 1.0 in room B and 1.0 in room A for Scenario 11;
- 1.7 in room B and 1.4 in room A for Scenario 12.

This means that when the clothing thermal resistance is lower, the PMV distribution is wider. The same trend could also be seen for the other scenarios in the same figure. Among the analyzed scenarios (9–12), for the north-facing room (room B), the scenario for which

50% of observations were inside the comfort range was Scenario 9, while in the south-facing one (room A), it was Scenario 10.

In the scenarios describing bedridden patients with short pajamas during the nighttime in both rooms (Scenarios 13–16), Scenario 13 had the best distribution with a null mean equal to the median. Scenarios 15 and 16 were characterized by 50% of the PMV values lower than −0.5.

In the daytime, the distributions had an IQR lower than the ones during the nighttime. For bedridden patients, null values of the mean and median, with 50% of data inside the comfort range, were achieved by long pajamas with 48.0% of bedsheets (Scenario 27) or short pajamas with a 67.0% rate (Scenario 30), both in rooms A and B.

For the scenarios out of bed, the comfort was achieved with heavy clothing: long pajamas in Scenarios 36 and 38 and short pajamas with a dressing gown in Scenario 41.

In general, for scenarios with bedridden patients, high levels of coverage (94.1% and 67.0%) showed a reduced PMV variation in mean, median, $Q_1$, and $Q_3$ of ~0.1, passing from room B to room A. This value was ~0.2 for the configuration with low coverage levels of 48.0% and 23.3%. The IQR had similar values for the same scenario in the two rooms.

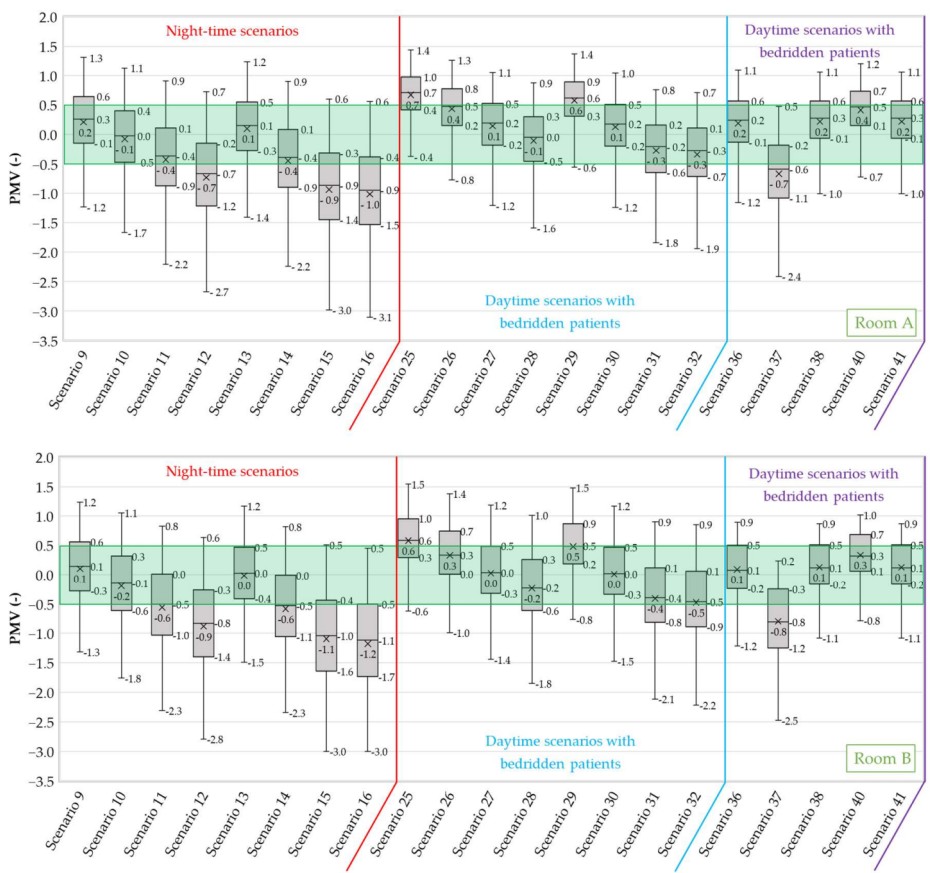

**Figure 21.** PMV distributions of the different scenarios in spring period.

In Figure 22, the results of the adaptive comfort model are shown. $T_{pma}$ is the average outdoor temperature calculated with the mean daily temperatures of the 30 days preceding the one in question. The points are the $T_o$ values for each hour, referring to different seasons. For instance, in spring, the results are shown by green points ($T_{o\ spring}$). It can be seen that for most of the observed hours, the $T_o$ values during the spring period were within the limit imposed by the legislation. Only in some cases were the values outside the minimum and maximum range: their percentages are reported in Table 6. The comparison between the conditions in rooms B and A showed double the points above the $T_{up\ limit}$ and half of the points below the limit $T_{low\ limit}$ in the south exposed room (room A) with respect to the

north exposed room (room B). Data concerning the summer period ($T_{o\ summer}$) followed a wavy pattern; this trend is the result of the combination of the outdoor conditions, the building thermal response, and HVAC operation.

### 4.2.2. Summer (15 June–15 September)

In the summer period, the mean value of $T_o$ over the whole period was 26.3 °C, and the RH was 65%. They were within the limits suggested by ISO 7730: $T_o \approx 24.5$ °C $\pm 1.5$ °C and RH 30–70%.

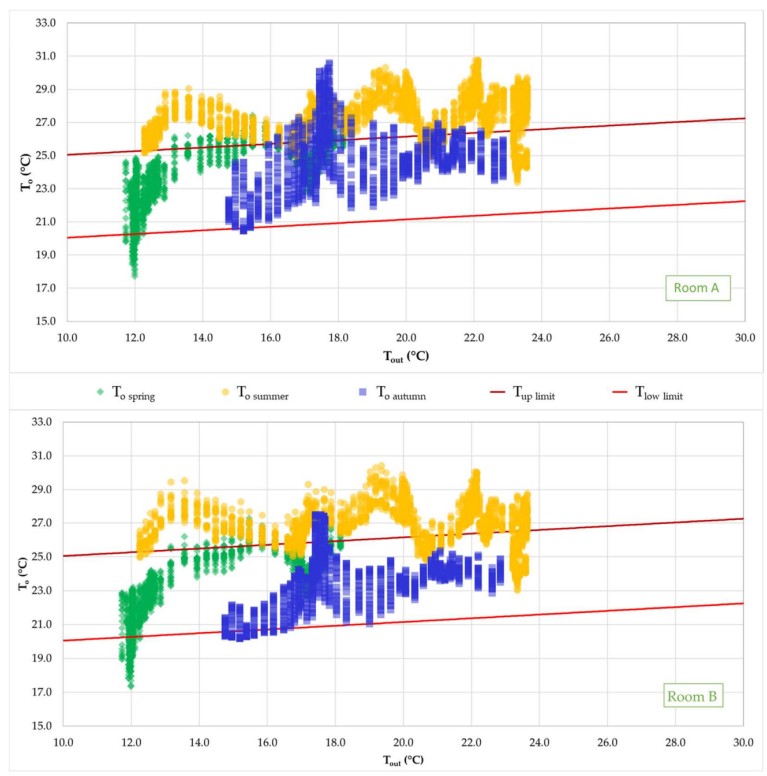

**Figure 22.** Adaptive model in spring, summer, and autumn periods.

**Table 6.** Percentage of data out of adaptive model bounds.

| Season | Observation over $T_{up\ limit}$ | | Observation under $T_{low\ limit}$ | |
|---|---|---|---|---|
| | Room B | Room A | Room B | Room A |
| Spring | 6.6% | 13.8% | 11.5% | 6.5% |
| Summer | 80.2% | 87.1% | - | - |
| Autumn | 7.8% | 25.7% | 6.2% | 0.6% |

In Figure 23, the box and whiskers graph for the summer period is reported. In regard to the nighttime scenarios (9–16), the distributions were nearly symmetrical for both rooms, and the median decreased with the decrease of bedsheet percentage of coverage. Moreover, the IQR was almost the same in room B as in room A. It is noted that more than 50% of all scenarios were greater than PMV = +0.5. Only the conditions of short pajamas and minimum percentage of bedsheet coverage showed a mean and median inside the comfort range for room B and slightly higher than +0.5 for room A.

Considering the daytime, for each scenario with a bedridden patient (scenarios 25–32), more than 50% of the data described an uncomfortable thermal sensation, between "slightly warm" to "warm". Furthermore, in this case, the median values of PMV decreased with the decrease of bedsheet coverage. In the case of the patient sitting or standing up, in room

B, only Scenario 39 had 50% of the data inside the comfort zone. The same scenario in room A showed a $Q_1$ inside the comfort zone.

Unlike nighttime conditions, all daytime scenarios (25–32 and 36–41) in room A had wider distributions than in room B. They had almost the same minimum and $Q_1$ values, but higher maximum and $Q_3$ values, and a difference of about 0.3 between the two rooms. During the daytime in summer, the contribution of solar radiation to the inner microclimatic conditions was strongly felt in the southern exposure room.

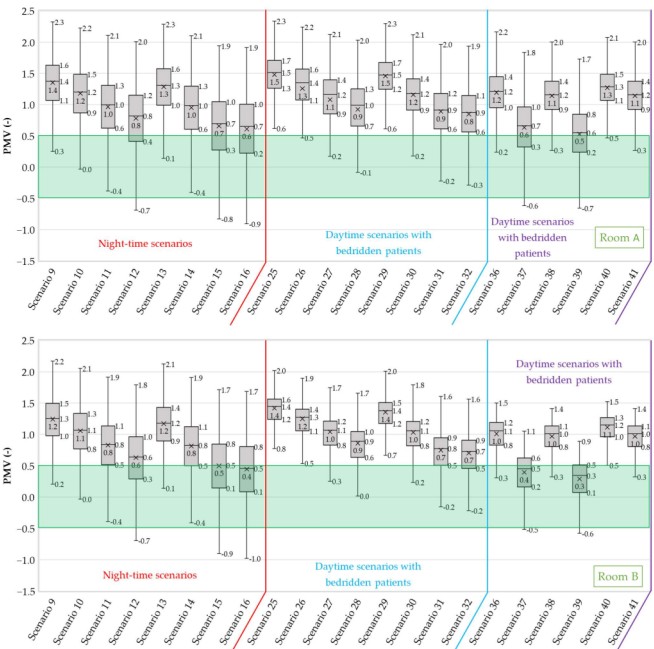

**Figure 23.** PMV distributions of the different scenarios in summer period.

Figure 22 is in accordance with the results of the PMV, as most of the $T_{o\ summer}$ values fell over the $T_{up\ limit}$. Specifically, this situation occurred in more than 80% of the hours for both rooms (Table 6).

### 4.2.3. Autumn (16 September–31 October)

The box and whiskers graphs for the autumn season are shown in Figure 24. During the night hours (scenarios 1–8), there were no comfortable conditions both for long or short pajamas if the maximum coverage level was used (Scenarios 1 and 5). On the other hand, the best distribution in the case of long pajamas was given by Scenario 3, both for rooms B and A; for short pajamas, it was Scenario 7 in room A and Scenario 6 in room B.

Regarding the daytime hours in which the patient was bedridden (Scenarios 17–24), satisfactory judgments regarding the internal microclimate were observed with minimum levels of coverage and long pajamas (Scenario 20) and medium levels of coverage and short pajamas (Scenario 23). For these distributions, in room B, more than 50% of data were inside the comfort range, while in room A, the $Q_3$ was greater than PMV = +0.5.

Finally, in the cases in which the patient was out of bed, the conditions for reaching the comfort zone were opposite in the south-facing room compared to the north-facing one. In room A, it was preferable to wear long pajamas (Scenarios 36 and 38). In room B, the comfort zone reached more than 50% of observations if a short pajama, with or without a dressing gown, was used (Scenarios 37 and 39).

As already observed in the previous time periods, the IQR of each scenario distribution increased with the decrease of coverage rate in autumn.

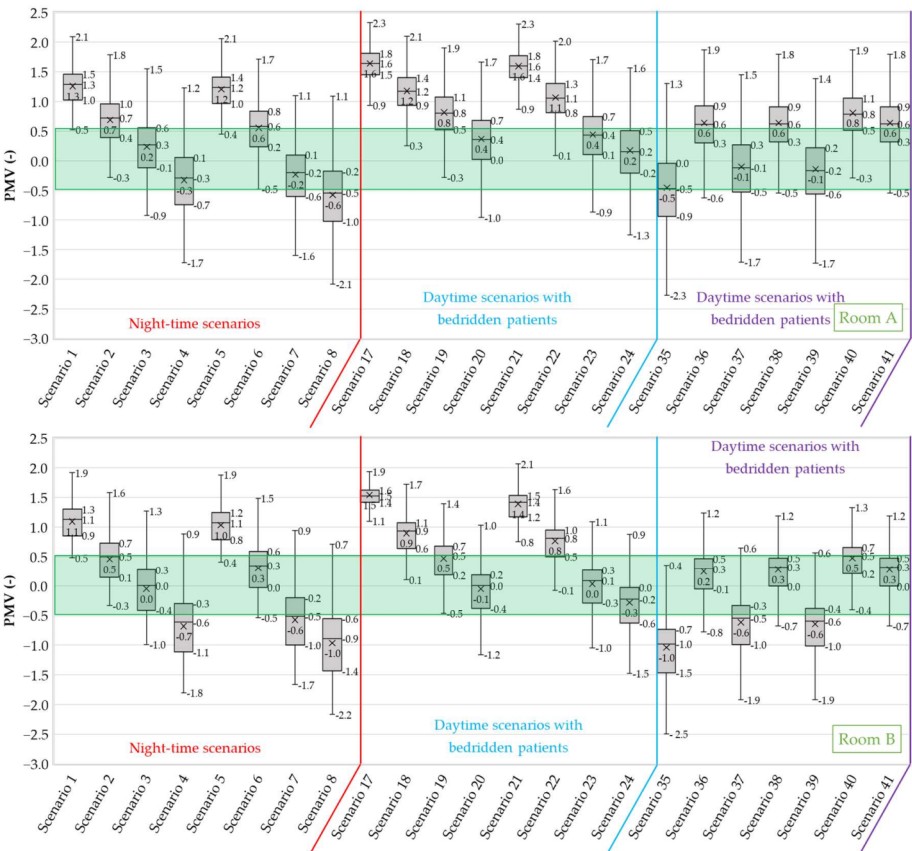

**Figure 24.** PMV distributions of the different scenarios in autumn period.

Considering the adaptive model applied to the autumn season, in Figure 22, the results are indicated by blue points ($T_{o\ autumn}$). In this case, the difference between the hygro-thermal conditions in the two rooms is evident. The percentage of $T_{o\ autumn}$ greater than $T_{up\ limit}$ passes from 8% in room B to 26% in room A (Table 6).

### 4.2.4. Winter (1 November–15 April)

In this period, the HVAC system was turned on. The average value of indoor $T_o$ was 20.5 °C, and RH was 50% in both rooms. Both are within the limits suggested by ISO 7730 for light activity, basically sedentary in winter: $T_o \approx 22$ °C $\pm 2$ °C and RH 30–70%. The box and whiskers graph in Figure 25 shows there was a lower IQR than all of the previous periods, with little total dispersed distributions: 0.5–1.2 in room B and 0.8–1.9 in room A. During the nighttime, the quilt with 67.0% of coverage and the long pajamas allowed the total PMV distribution inside the comfort range $\pm 0.5$ in both rooms. Meanwhile, the coverage rate lower than 67.0% had a thermal sensation of "slightly cool" or "cool".

During the day hours, when the patient was bedridden, Scenarios 18 and 19 were suitable. Furthermore, Scenario 40 (patient out of bed with the long pajamas and dressing gown) had the total distribution inside the comfort range, in room B, and the maximum value (PMV = 1.0) in room A.

In this case, like the daily analysis, it was not possible to make a comparison with the adaptive model as the average daily outdoor temperatures were, in most cases, less than 10.3 °C, and therefore, below the limit of applicability of the adaptive model.

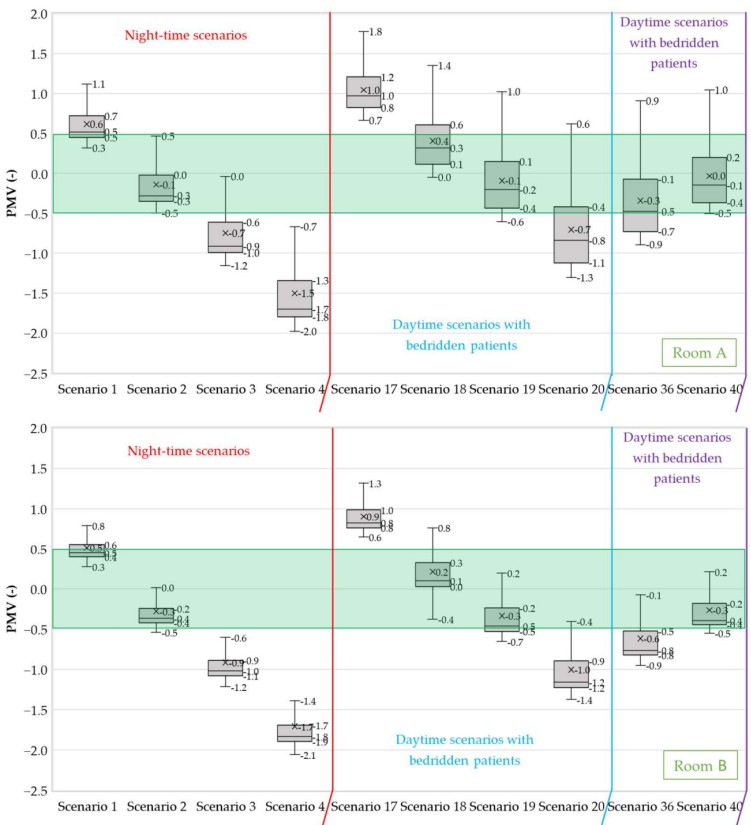

**Figure 25.** PMV distributions of the different scenarios in winter period.

## 5. Discussion

In this section, the Authors attempt to analyze the critical issues that appeared in the application of static or adaptive comfort models in hospitals.

First of all, it has been observed that when the HVAC system is turned on, in winter (1 November–15 April) and in summer (15 June–15 September), the indoor thermo-hygrometric parameters are stable and are within the limits suggested by ISO 7730 for light activity, basically sedentary. Therefore, the analyzed results in this paper could be considered representative of the thermo-hygrometric conditions of other hospitals designed following the international comfort standard. During the intermediate period (spring and autumn), the free-floating conditions have been presented and discussed.

In the winter period, the level of coverage greatly affected the perception of thermal comfort: if the right combination of quilt coverage and clothing is adopted, for all the hours of the analyzed period, the bedridden patient could be in the comfort range. It is not always true if the room is south facing due to the greater contribution of solar radiation that shows hours with a "slightly warm" sensation. It can also be deduced that if the internal conditions are those suitable for bedridden patients, a patient sitting out of bed is not in comfortable conditions.

Regarding the summer period, from both daily and seasonal results, various observations can be made. The first one is that in a hospital, the temperature range values conventionally adopted for a light activity, basically sedentary (such as residences or offices) patient, do not guarantee a dissatisfied percentage less than 10% in the case of bedridden patients. The additional resistance of the bed (always present) and the possible cover sheet determines the need to further reduce the indoor temperatures, despite the condition of the patient at rest, therefore, with a low metabolic level. For more than 50% of the observations in each scenario, the bedridden patient feels discomfort conditions during the daytime in summer, between "slightly warm" to "warm". Therefore, during the summer, plant operation may need to be rescheduled, for instance, by using two levels of set-points, one

for day and one for nighttime. Confirming this, it has been observed that in cases where the patient is out of bed, on the basis of the Fanger model, a positive judgment was attributed to the microclimate.

In the intermediate period, with the HVAC turned off, the indoor microclimate conditions were floating, so the judgments against it were widely dispersed and could go from "slightly warm" to "slightly cold" for the same scenario. Furthermore, different sensations towards the microclimate could be achieved if the orientation of the room varied, with the same window area and same clothing insulation level.

Considering the differences between the two models, first of all, it should be emphasized that in the static model, both the metabolic rate and thermal resistance of the clothing were accurately assessed, while the adaptive comfort model was based on ranges already defined for these factors. During the summer period, the analysis of the adaptive model was in accordance with the results of the static model, suggesting that the indoor set-point of the air temperature must be redefined (reducing) in order to make the indoor environment more comfortable.

On the other hand, during the intermediate periods, the application of the two models leads to discordant results. For instance, in spring, the static model achieved thermal neutrality for 100% of the hours of the analyzed period by varying the insulation of clothing or the level of coverage when the patient is in bed. Meanwhile, according to the adaptive model, about 20% of the total hours showed uncomfortable conditions. The fundamental limit of the adaptive model was not considering clothing resistances as high as in the case of a bedridden patient. Similar conclusions could be written for the autumn period, characterized by a greater number of discomfort hours, above all in the south-facing room (27%).

Finally, it should be emphasized that the patient in a surgery ward, in addition to being confined to bed, is not able to modify the conditions of the context that surrounds them; consequently, the application of the static model was more appropriate. The evaluation of the PMV allowed us to take into account the type of clothing, the influence of the bed, and the metabolic rate, but it is completely detached from external climatic conditions. The analysis with an adaptive approach can provide discordant results with respect to the static model.

## 6. Conclusions and Future Developments

In this paper, the thermo-hygrometric conditions in ordinary hospitalization rooms of a surgery ward of the "Ferdinando Veneziale" Hospital are presented. The analysis was carried out by means of a calibrated energy numerical model in the EnergyPlus engine. Different characteristics and specific factors of the hospitalized patient that significantly influence the comfort indices were taken into account. In total, 41 scenarios were defined and analyzed by means of two comfort models: static and adaptive.

The first distinction must be made between periods with HVAC systems on and off. When the HVAC is on, in winter and in summer, the indoor thermo-hygrometric parameters are stable and within the limits suggested by ISO 7730 for light activity, basically sedentary. Thus, the analyzed results in this paper could be considered as representative of the thermo-hygrometric conditions of other hospitals designed following the international comfort standard. During the intermediate period, the free-floating conditions of the results could be representative of other hospital wards in a Mediterranean climate.

The main results could be summarized as follows.

During the winter period:

- If the patient is bedridden, long pajamas with 67.0% quilt coverage can achieve neutral sensation for all hours of the days in which HVACs are turned on.
- An increase of PMV ($\approx$ +0.1) has been observed in the south-facing room with respect to the north-facing one.
- Thanks to the bed thermal insulation and different coverages, the conditions suggested by ISO 7730 for light activity in wintertime (operative temperature $\approx$ 22 °C $\pm$ 2 °C

and relative humidity 30–70%) seem appropriate for a bedridden patient in a surgery ward. Only during some hours (lower than 25% of the total period) was a thermal sensation of "slightly warm" observed in the room most exposed to solar radiation.

In summer:

- The thermal insulation of the bed system is a pejorative factor that, however, cannot be eliminated.
- The patient with a low coverage rate of bedsheets could create comfortable conditions more easily.
- For each possible combination of clothing-bed covering during the daytime, there were more than 50% of hours that had a thermal sensation between "slightly warm" and "warm".
- The analysis of the adaptive model for the summer season was in accordance with the results of the static model (more than 80% of the hours are greater than the upper limit of 90% of the acceptability category).
- The microclimate conditions suggested by ISO 7730 ($T_o \approx 24.5\,^{\circ}C \pm 1.5\,^{\circ}C$ and RH 30–70%) were not suitable for the hospital environment.

During autumn and spring:

- The thermal sensation with the same clothing-bed covering toward the microclimate can vary greatly within the same day, between "slightly warm" to "slightly cool".
- In the cases in which the patient is out of bed, the optimal clothing-bed covering conditions are opposite in the south-facing room compared to the north-facing one. In south rooms, it is preferable to wear long pajamas, while, in north rooms, the thermo-hygrometric comfort is reached in short pajamas, with or without a dressing gown.
- The application of the two models led to discordant results. For instance, in spring, if a static model is used, the variation of clothing or the level of bed coverage could be successful in achieving thermal neutrality. While according to the adaptive model, about 20% of the total hours are in not comfortable conditions. The fundamental limit of the adaptive model is not to consider clothing resistances as high as in the case of a bedridden patient.
- Similar conclusions could be written for the autumn period, characterized by a greater number of discomfort hours, above all, in the south-facing room (27%).

The results of the analysis carried out confirm that the heterogeneity of conditions that can occur inside a hospital room in terms of patient behavior (out of bed or in bed with different degrees of coverage) and clothing make it difficult to apply the usual comfort models.

Thanks to the large number of analyzed scenarios, this study could be the starting point for defining a standard of the thermal environmental conditions tailored to a hospital. A monitoring plan that includes in-field measurements and questionnaires has been defined and will be put in place after the COVID-19 pandemic. The questionnaires will be administered differently to patients and managers, and they will regard: general information on the state of the people (behavior and preferences) and judgments about air temperature, relative humidity, air speed, and its quality. At the same time, in some representative rooms, microclimate control units will monitor the same parameters. In this way, more information about the state of patients, pre or post-operative, and their pathologies, drug intake, and physical conditions will be involved in the study. Moreover, it could also be studied whether the use of common thermal comfort indices, conventionally applied in other buildings, are suitable to represent and understand patient comfort in hospitals.

**Author Contributions:** Conceptualization, S.R., F.T. and G.P.V.; methodology, S.R., F.T. and G.P.V.; software, S.R., F.T. and G.P.V.; validation, S.R., F.T. and G.P.V.; writing—original draft preparation, S.R., F.T. and G.P.V.; writing—review and editing, S.R., F.T. and G.P.V.; supervision, S.R., F.T. and G.P.V.; All authors have read and agreed to the published version of the manuscript.

**Funding:** The authors gratefully would like to thank the financial support from the "PRIN 2017-SUSTAIN/ABLE–SimultaneoUs Structural And energetic reNovAtion of BuiLdings through innovative solutions–MIUR, DD N. 240 DEL 14.02.2019, CUP F84I19001170001".

**Institutional Review Board Statement:** Not applicable.

**Informed Consent Statement:** Not applicable.

**Conflicts of Interest:** The authors declare no conflict of interest.

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
