# Peer review of "Evaluation of Comfort Models Considering the Peculiarities of Hospitalization: Bedding, Clothing and Reduced Activity of Patients"

_buildings, doi:10.3390/buildings12030343_

Round 1

Reviewer 1 Report

The study aimed to evaluate the static and adaptive comfort models considering the clothing, bedding and activity level of patients in a hospital in Southern Italy. The study mentioned 41 scenarios being tested. However, the results don't present all these scenarios. The study lacks monitoring and calibration of the existing building. Therefore, the results can not be verified. 

You may find detailed comments below: 

-Line 60: Do you mean exclusive?

-Line 168: Write " Five". Don't start a new sentence with a number.

-Literature review: The hospitals studied in LR followed the same method or also had the same facilities? Construction, materials, conditions? The focus of LR is mainly on studies that used the PMV method rather than having a detailed methodological and analysis approach.

-The study surely has some Limitations, it could be an exclusion of type of disease, patient profile, weather data, monitored data, HVAC systems, etc. Which are not clearly stated.

-Lines 213-218: How you can relate/ have considered this in your study?

-Line 226-230:  How you can relate/ have considered this in your study?

-Line 288: Add a picture of the model, created in DesignBuilder. Include the input data in annexures. Considering the real buildings and their operations.

-Elaborate on the selected rooms, floor/level, where the HVAC units/ventilation ducts are located (duct, inlet, radiator).

-Line 255: Add a code of climate classification, i.e. Cfa, BSk etc.

-Line 290-292: the setpoint temperatures are based on actual practice/ values in the existing building? Based on any code or standard? Or Used for simulation purposes only?

-Line 311: Thermophysical characteristics aren't really defined. But U-Values of the envelope is mentioned. Thermophysical characteristics would include the layers of construction, materials, their densities, specific heat capacity, thermal conductivity, thickness etc.

-Any monitoring was done? How the input data was collected for simulation?

-It is not clear if the model is calibrated. In case, it was then provide with calibration data and details. 

-There is no logical representation of results. the results section doesn't enlighten the reader, on how the results are presented. 

-Not all scenarios are covered in the results section. Why?

-Table 1: How if we relate the scenarios with a specific season/ representative days.  

-Referencing style is incorrect. A few names are written in full, or the first name is written first.  All references should be rechecked. 

-Enlist the important contribution and results in bullets.

In case, the results are produced without calibration of the model then the findings have no or less significance. The paper can be not be accepted in its current form without addressing this issue. Otherwise, include a limitations section enlisting all limitations of the study. 

Reviewer 2 Report

In this paper a study of the evaluation of comfort models considering the peculiarities of hospitalization, mainly bedding, clothing and reduced activity of patients, was made. Before being published, I suggest some improvements in the presentation and in the content.

More details about the numerical model applied in this study should be added, as equations, constants, and other details.

More details about the inputs should be added.

The calibration of the numerical model should be improved. More details should be added.

More information should be presented about the different simulations. Use a table, as example, to introduce the main inputs and methodologies uses in each simulation.

The adaptive model, used in the paper to evaluate the comfort conditions, should be improved.

More details about the future works should be added.

Reviewer 3 Report

This is a detailed simulation study of comfort of patients in a real Italian hospital, using standard comfort theories and a very wide range of patient clothing/activity scenarios across 4 seasons.  Detailed results are presented graphically, mainly summarised by day or season, although an example plot of one day to show the process in more detail with 24 hourly values would be useful. The application to hospitals which are not much researched for comfort, and range of scenarios and times of day, season are its main strength; its obvious limitation is that it is entirely based on theoretical comfort, while there have been many studies showing how this can diverge from reality - some hinted at in the review.  It is good to see that an empirical study will follow. 

It is very well written and formatted, but a number of minor comments have been added to the attached MS as corrections or suggestions; and a suggestion to reconsider the structure of Discussion and Conclusions.

Overall this is a useful paper but needs to be validated with empirical studies.

Round 2

Reviewer 1 Report

The authors have made all necessary changes and corrections. The quality of the paper is improved including its methodology and results section. 

Some spelling, words and sentences can be corrected for better understanding. 

For example, Fig. 2-b. Request for natural gas. Do you mean gas consumption or demand? 

Author Response

Reviewer #1: The authors have made all necessary changes and corrections. The quality of the paper is improved including its methodology and results section. 

Some spelling, words and sentences can be corrected for better understanding. 

The entire manuscript has been reviewed making some changes to words or sentences where it seemed more appropriate.

For example, Fig. 2-b. Request for natural gas. Do you mean gas consumption or demand?

Fig. 2 has been modified. Sub-figure b represents gas consumption.

Reviewer 2 Report

In the actual version, in general, all suggestions given by the reviewer was implemented.

Author Response

Thank you for the review.